# A genetic screen reveals a key role for Reg1 in 2-deoxyglucose sensing and yeast AMPK inhibition

Alberto Ballin[1], Véronique Albanèse [1], Samia Miled[1], Véronique Legros[2], Guillaume Chevreux[2], Agathe Verraes[1], Anne Friedrich[3], Sébastien Léon[1]*

1 Université Paris Cité, CNRS, Institut Jacques Monod, Paris, France, 2 ProteoSeine Core Facility, Université Paris Cité, CNRS, Institut Jacques Monod Paris, France, 3 Université de Strasbourg, CNRS, GMGM UMR 7156, Strasbourg, France

* sebastien.leon@ijm.fr

## Abstract

The yeast *Saccharomyces cerevisiae* thrives in sugar-rich environments by rapidly consuming glucose and favoring alcoholic fermentation. This strategy is tightly regulated by the glucose repression pathway, which prevents the expression of genes required for the utilization of alternative carbon source. Central to this regulatory network is the yeast ortholog of the heterotrimeric 5′AMP-activated protein kinase (AMPK), which adjusts gene expression in response to glucose availability. The activity of the yeast AMPK complex is primarily regulated by the phosphorylation state of its catalytic subunit Snf1, a process orchestrated by a balance between upstream kinases and phosphatases. Among the latter, the Protein Phosphatase 1 (PP1) complex Reg1/Glc7 plays a critical role in inhibiting Snf1 activity under glucose-rich conditions. Despite its importance, the precise mechanism by which glucose availability leads to Snf1 inhibition remains incompletely understood. Evidence suggests that hexokinase 2 (Hxk2) participates in this pathway, potentially coupling the early steps of glucose metabolism to Snf1 signaling. Notably, the toxic glucose analog 2-deoxyglucose (2DG)- which is phosphorylated by Hxk2 but not further metabolized-mimics glucose in its ability to repress Snf1, implicating glucose or 2DG phosphorylation as a key regulatory signal. Additionally, yeast AMPK activity correlates with 2DG resistance through mechanisms that are incompletely described. In this study, we performed a large-scale 2DG-resistance genetic screen to explore both the molecular basis of 2DG resistance and AMPK regulation in yeast. The identified mutations confer resistance either by reducing 2DG phosphorylation (e.g., mutations in *HXK2*) or by enhancing constitutive Snf1 activity, via gain-of-function alleles in AMPK subunits or loss-of-function mutations in *REG1* and *GLC7*. We also describe a novel series of *REG1* missense mutations, including *reg1-W165G*, that maintain basal, glucose-regulated Snf1 activity but fail to mediate 2DG-induced Snf1 inhibition. These findings

---

**Data availability statement:** The data used to generate the figures are available S1 Data. The MS data is available on the PRIDE repository under accession number PXD068555.

**Funding:** AB was supported by a PhD fellowship salary from the French Ministry for Education and Research and from the Graduate School "Ecole Universitaire de Recherche - Génétique et Epigénétique Nouvelle Ecole" (EUR GENE, ANR-17-EURE-0013, www.anr.fr). This work was supported by a grant from the Agence Nationale pour la Recherche ("AMPKILL", ANR-23-CE13-0012-01, www.anr.fr) to SL. The funders had no role in study design, data collection and analysis, decision to publish, or preparation of the manuscript.

**Competing interests:** The authors have declared that no competing interests exist.

position Reg1 as a central mediator in glucose sensing, possibly by sensing 2DG-derived -and by extension, glucose-derived- metabolites.

## Author's summary

Yeast such as Saccharomyces cerevisiae thrive in sugar-rich environments by rapidly consuming glucose and converting it to ethanol, even when oxygen is present. This strategy relies on a pathway named "glucose repression pathway", which blocks the use of alternative carbon sources. At the core of this pathway is Snf1, the yeast equivalent of mammalian AMPK, whose activity is regulated by upstream kinases and the Reg1/Glc7 phosphatase complex. Despite extensive research, how glucose metabolism inhibits Snf1 remains unresolved.

A useful tool to probe this pathway is the toxic sugar mimic 2-deoxyglucose (2DG), which, once phosphorylated by hexokinase 2, mimics glucose in shutting down Snf1, although it cannot be metabolized further. Here, we performed a large-scale screen for yeast mutants resistant to 2DG. Resistance arose either from reduced 2DG phosphorylation or from mutations that keep Snf1 active despite glucose or 2DG. Crucially, we identified novel Reg1 missense mutations that preserve glucose regulation but fail to mediate 2DG-dependent Snf1 inhibition. These findings reveal Reg1 as a central mediator of glucose sensing, and suggest it may directly interpret signals from early sugar metabolism—a previously unrecognized role that reshapes our understanding of how yeast couples nutrient detection to metabolic control.

## Introduction

The yeast *Saccharomyces cerevisiae* has evolved in sugar-rich niches and is able to compete with other microbial species through an adapted metabolism based on a fast and efficient glucose consumption coupled to alcoholic fermentation, even in the presence of oxygen [1]. Consequently, the ability to detect available sugars is key to survival, and several signaling pathways allow yeast to adapt its metabolism with respect to changes in the carbon source available [2]. In particular, *S. cerevisiae* has a clear preference towards glucose: the presence of glucose prevents the expression of genes involved in the use of alternative sugars as well as in respiration, also known as the "glucose repression pathway" [3,4].

Decades of research led to the identification of several actors of this pathway (presented in Fig 1A). Early genetic screens isolated mutants unable to use various carbons sources such as glycerol, ethanol or sucrose [5–7]. These mutants failed to induce the expression of various enzymes involved in the metabolism of these carbon sources. These mutants were allelic and mutated in a gene named *SNF1* (sucrose non-fermenting 1). Snf1 turned out to be the yeast orthologue of the

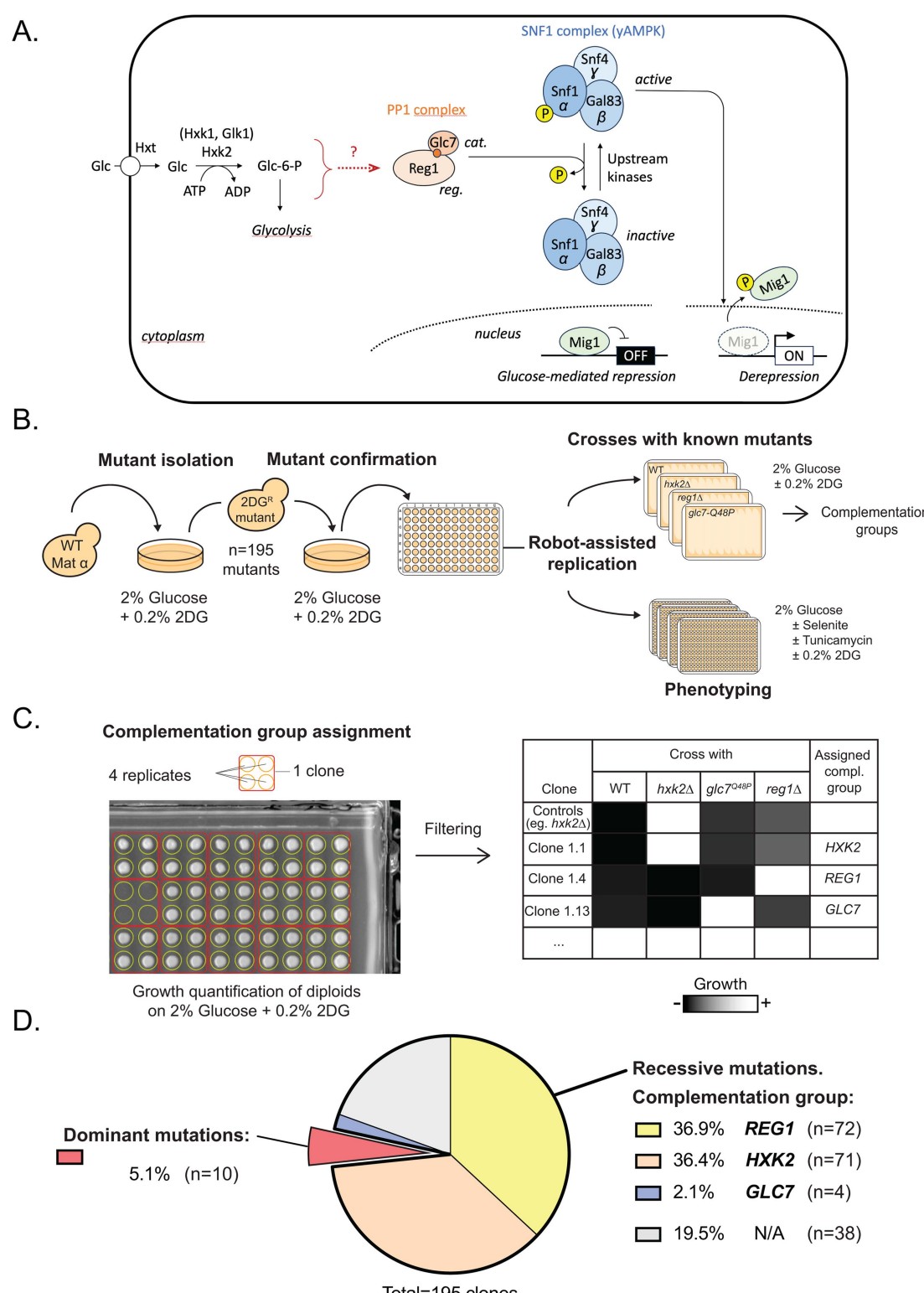

**Fig 1. Genetic screen for isolation of 2DG mutants and genetic analysis. A.** Working model for the regulation of yeast SNF1 activity by glucose availability. Glucose enters the cell through hexose transporters (Hxt) and is phosphorylated to glucose-6-phosphate by the main hexokinase isoform Hxk2, as well as by Hxk1 and Glk1, initiating glycolysis. A yet unidentified metabolic signal generated during these early steps ("?") promotes the

dephosphorylation of Snf1, the catalytic (α) subunit of the yeast AMPK complex, thereby maintaining Snf1 in an inactive state. This inactivation requires the PP1 phosphatase, composed of the regulatory subunit Reg1 and the catalytic subunit Glc7. Snf1 activity determines the phosphorylation status of Mig1, a transcriptional repressor responsible for glucose-dependent repression of genes involved in respiration and the utilization of alternative carbon sources. When glucose is scarce, PP1 no longer acts on Snf1, allowing its activation. Active Snf1 phosphorylates Mig1, relieving glucose repression and enabling expression of genes required for the use of non-glucose substrates. **B.** Schematic of the experimental setting used for the isolation of spontaneous 2DG-resistant mutants. WT cells were plated on selective medium containing 2% glucose supplemented with 0.2% 2DG. 195 colonies (representing resistant mutants) were picked for further studies. Their resistance phenotype was confirmed by an additional selection on 2DG. Mutants were then subject to complementation group tests and to growth assay in various conditions. The mutants were spotted using a robot for high-throughput manipulation of microorganisms. **C.** Schematic of the workflow used for the quantification of the diploids' growth and their assignment to a complementation group. Colony size after 3 days of growth was quantified using ImageJ and compared to control strains. A threshold was chosen to affiliate mutants as part of a complementation group (see Material and Methods). The data is shown in S1 Table, representing growth of diploids derived from crosses of each mutant with control strains (*hxk2Δ, glc7-Q48P, reg1Δ*). **D.** Pie-chart showing mutant distribution within the complementation groups and the proportion of unassigned and dominant mutants.

catalytic (α) subunit of the mammalian AMPK (5'-AMP-activated kinase) complex [8]. AMPK is a heterotrimeric Ser/Thr protein kinase composed of three subunits (α, β, ɣ) which in mammalian cells allows cells to cope with energy stress by attenuating energy-consuming reactions (e.g., protein or fatty acid synthesis) while favoring energy production (e.g., glucose uptake and glycolysis, fatty acid oxidation or mitochondrial biogenesis) [9]. This discovery highlighted the conservation of energy-sensing mechanisms across eukaryotes, and a specialization of the yeast AMPK complex in the preference of yeast for glucose. Indeed, a mutant with a similar phenotype as *snf1* is mutated in *SNF4* [10,11], which encodes the ɣ-subunit of the complex. Other genes encoding three partially redundant β-subunits were also identified by genetic screens or protein interaction studies [12–15]. Altogether, the yeast AMPK complex (here named 'SNF1') is required for the expression of genes involved in the metabolism of non-glucose carbon sources upon glucose depletion.

The effect of the SNF1 complex on transcriptional regulation is mainly mediated by the phosphorylation of the transcriptional repressor Mig1 [16,17]. Mig1 phosphorylation prevents its interaction with the general repressor complex Cyc8-Tup1 [18] and leads to Mig1 cytosolic redistribution [19,20]. Since Mig1 function is inhibited by SNF1, loss of function of *MIG1* cause the opposite phenotype as mutations in AMPK-encoding genes, i.e., a constitutive expression of glucose-repressed genes [21–23], and so do mutations in *CYC8* or *TUP1* [22,24].

These data implied that SNF1 activity is increased when glucose is absent, in order to relieve gene repression through Mig1 phosphorylation. This brought the question of how SNF1 activity is regulated by glucose availability. The activity of kinases of the AMPK family is regulated by phosphorylation on a regulatory region of the α-subunit named "T-loop". This phosphorylation is increased in conditions of glucose scarcity [25] and is accompanied by changes in Snf1 conformation and molecular interactions between subunits of the SNF1 complex [26,27], altogether leading to SNF1 activation. Three upstream kinases can mediate Snf1 phosphorylation [28–30]. However, there is no evidence that glucose availability regulates their activity, supporting a model in which Snf1 phosphorylation is constitutive, while its dephosphorylation is controlled by glucose levels [31].

The identity of this phosphatase was again revealed through genetic screens aimed at identifying mutants defective for glucose repression. These studies used the glucose analogue 2-deoxyglucose (2DG), which mimics glucose in the glucose repression pathway but does not provide energy and cannot be used for growth [32,33]. Thus, in presence of 2DG, enzymes required for the use of alternative carbon sources are not expressed, but mutants that have lost the glucose-mediated repression of genes are able to use these sources, such as sucrose [5] [reviewed in 34]. Several mutants were isolated, some of which were mutated in *GLC7*, encoding the catalytic subunit of protein phosphatase 1 (PP1) [35] or in the *REG1* gene, encoding a regulatory subunit of the same complex [5,36,37]. Reg1 acts in concert with Glc7 to inhibit Snf1 activity [37,38] and accordingly, Snf1 is constitutively phosphorylated in cells lacking Reg1 [25,39]. Moreover, data suggest that Reg1/Glc7 can also dephosphorylate Snf1 substrates, such as the repressor Mig1 [31,40]. Altogether, these findings support a model in which the presence of glucose inhibits SNF1 activity by enhancing the activity of the Reg1/Glc7 phosphatase complex towards Snf1 or its substrates.

The precise nature of the signal by which glucose activates the PP1-mediated dephosphorylation of Snf1 remains elusive. Mutations in *HXK2,* encoding the major hexokinase isoform, cause a constitutive expression of several glucose-repressed enzymes, suggesting that the glucose-repression pathway is tightly connected to glucose metabolism and/or that Hxk2 is both a metabolic enzyme and a signaling protein [41–44]. This was supported by the observation that basal phosphorylation of Snf1 in glucose-containing medium is partially elevated in the *hxk2Δ* mutant [45,46], as well as that of Mig1 [47,48]. On the other hand, it was reported that deletion of *HXK2* does not lead to major transcriptional changes, which is difficult to reconcile with previous data [49]. Therefore, to date, the function of Hxk2 and the molecular nature of the signal triggering the glucose-induced dephosphorylation of Snf1 by Reg1/Glc7 remains unknown.

We previously showed that 2DG addition to glucose-grown cells causes a decrease in the basal phosphorylation of Snf1 and its substrates [50]. This suggested that 2DG treatment mimics a situation of excess glucose, despite its energy-depleting effects. This response was not observed upon deletion of *HXK2*, which we proposed to be the main 2DG-phosphorylating enzyme *in vivo*. Thus, 2DG phosphorylation, i.e., 2DG-6-phosphate itself or a derived metabolite, is central to trigger Snf1 dephosphorylation. This is supported by the observation that overexpression of 2-DG-6-phosphatases inhibits the effect of 2DG in the glucose repression pathway [50,51]. It also demonstrated that 2DG can be used as a proxy to probe SNF1 inhibition at the signaling level, independently of the energetic effects obtained when using glucose.

As mentioned above, genetic screens aimed at identifying mutants that are resistant to 2DG allowed to identify actors involved in the glucose-regulated activity of the SNF1 complex [reviewed in [34]]. In this study, we performed a large screen to identify spontaneous 2DG-resistant mutants, with the aim to gain further insights into the mechanisms of 2DG resistance as well as to dissect the molecular mechanisms in charge of 2DG-induced Snf1 inhibition by isolating new tools for mechanistic studies. We report several 2DG-resistance strategies. A first one originates from mutations in *HXK2*, likely through a reduced phosphorylation of 2DG and thus a lower toxicity, confirming previously published results [45,46,52–54]. The second one is caused by an increased basal SNF1 activity, either through gain-of-function mutations in all three subunits of AMPK (*SNF1, SNF4, GAL83*) as previously reported [55] or loss-of-function mutation in the PP1 phosphatase genes, *REG1* and *GLC7* [45,46,52,53]. Globally, this second class of mutations relies on a constitutive SNF1 activity to cope with 2DG toxicity, probably through detoxification mechanisms involving 2DG-6-phosphatases and/or glucose transport activity [50,53,56]. Additionally, we report a series of mutations caused by missense mutations in various conserved residues of Reg1. These were different from the abovementioned loss-of-function *reg1* mutants as they did not cause increased basal SNF1 signaling, suggesting a different resistance strategy. This is further supported by the observation that these mutants accumulate the toxic by-product of 2DG metabolism, 2DG-6-phosphate, whereas *reg1Δ* cells do not. We characterized several of these point mutants and one in particular, Reg1-W165G, which appeared insensitive to 2DG and thus did not display any of the dephosphorylation events normally observed upon 2DG treatment (Snf1 and its targets Mig1 and Rod1). Whereas 2DG treatment led to an increased interaction between Reg1 and the SNF1 complex, this did not occur for Reg1-W165G, suggesting that this residue is key for 2DG sensing and induced recruitment of Reg1 onto SNF1. These findings allow us to propose a model whereby Reg1 is at the core of the glucose sensing mechanism and is likely involved in the direct sensing of the metabolic signal originating from the early steps of glycolysis (Fig 1A).

## Results

### Isolation of spontaneous 2DG-resistant mutants and complementation analysis

To gain insights into how 2DG is sensed and how it regulates AMPK activity, we performed a genetic screen to isolate 2DG-resistant mutants in glucose-containing medium. We spread ca. 90 x 10^6 cells on 10 plates of glucose (2%)-containing complete synthetic medium supplemented with 0.2% 2DG, a concentration that is toxic to WT cells [52] (Fig 1B). About 1400 resistant clones spontaneously arose after 7 days. We further selected 195 mutants of various colony sizes (large, medium, and small) to increase the diversity in the mutations isolated. The overall workflow is presented in S1 Fig.

Based on previous genetic screens performed in these conditions [46,53] we expected to identify (i) mutants in *HXK2*, which encodes the main isoform of hexokinase which is mainly responsible for 2DG phosphorylation [50], (ii) mutants in *REG1* and *GLC7* which negatively regulate AMPK, whose activity correlates with 2DG resistance [46,53] as well as (iii) dominant, gain-of-function mutations in AMPK subunits [46,55]. Therefore, the 2DG-resistant (haploid) clones were subjected to complementation group analysis by crossing them with WT (to establish dominance/recessivity) or mutants in *HXK2, REG1, or GLC7* and measuring colony size of the obtained diploids on 2DG medium after 3 days of growth (Fig 1C). Strains handling was automated (RoToR robot, Singer) allowing reproducible replication in quadruplicate and thus enabling growth quantification through colony size measurement in ImageJ. Based on these data, we were able to assign 157 mutants (81%) to a single category - either dominant or belonging to the *HXK2, REG1,* or *GLC7* complementation groups (Fig 1D and S1 Table).

As reported in previous studies, the majority of 2DG-resistant mutants were in the *HXK2* (71 mutants) and *REG1* (72 mutants) complementation group (36% and 37% of all selected mutants, respectively). Targeted sequencing of *HXK2* from 5 random mutants belonging to the *HXK2* category confirmed either missense or nonsense mutations in the coding sequence (S1 Table). Four mutants belonged to the *GLC7* complementation group, a category that will be further scrutinized later in this study. Finally, ten mutants appeared as dominant. We hypothesized that these mutants carried mutations in various subunits of the heterotrimeric AMPK complex, since gain-of-function mutations were previously isolated in *SNF1, SNF4* and *GAL83,* respectively encoding the α, ɣ and the major isoform of the β subunits of AMPK [55,57–59]. Targeted or whole-genome resequencing of some of these mutants revealed mutations in either of these genes, as discussed in the next section.

In addition, 38 mutants could not be ascribed to the studied complementation groups. Several of them were tested for other genes whose deletion was already known to cause 2DG resistance, either by targeted sequencing or whole-genome resequencing. One clone (#2.22) carried a missense mutation in *ROD1*, encoding a glucose-regulated arrestin involved in endocytosis and whose deletion causes 2DG resistance [50,52]. The mutation (L554*) is predicted to truncate the C-terminal region of the protein which contains interaction sites with the ubiquitin ligase Rsp5, required for Rod1 function [56,60,61]. Clone #4.17 carried a mutation causing a truncation of the Cyc8 protein (W240*), extending previous results regarding its involvement in glucose repression [62,63] and 2DG resistance [53]. We also identified a mutation in *TPS2* (clone #5.21), encoding the phosphatase subunit of the trehalose-6-P synthase/phosphatase complex [64] whose deletion was also previously reported to cause 2DG resistance [52]. The identified mutation causes a single substitution (R619C) in the active site of the HAD-like phosphatase domain, which would presumably alter its activity, based on structural studies [65]. Other mutants carried mutations that could not be linked to 2DG resistance *a priori*. In particular, 2 clones carried the same missense mutation in *DCK1*, encoding a regulator of the Rho5 GTPase [66], causing a single residue substitution (T592S). Finally, two identical mutations were found in the gene *FYV10/GID9* (mutation S388L), encoding a protein involved in the glucose-regulated, proteasome-dependent degradation of gluconeogenic enzymes [67]. However, we did not confirm that these mutations are causative of the 2DG resistance phenotype at this stage.

### New gain-of-function alleles of AMPK causing a constitutive AMPK activity and 2DG resistance

Dominant mutations in genes encoding AMPK targeted either the *SNF1*, *SNF4* or *GAL83* genes (Fig 2A) and structural considerations are presented below. First, to confirm that the identified mutations are sufficient for 2DG resistance, we cloned the mutant genes in a plasmid allowing their expression under the control of their own promoter as a fusion with mCherry. This tag did not abrogate the function of the WT proteins as determined by their ability to complement the corresponding mutant strains for growth on galactose as a carbon source (Fig 2B). Moreover, the isolated mutations in *SNF1*, *GAL83* or *SNF4* did not significantly affect the expression level of the corresponding proteins (Fig 2C). These mutants were then expressed in WT cells which were then spotted on glucose-containing medium containing 0.2% 2DG. Contrary to the expression of the corresponding WT proteins, the expression of the *SNF1, SNF4* or *GAL83* mutants was

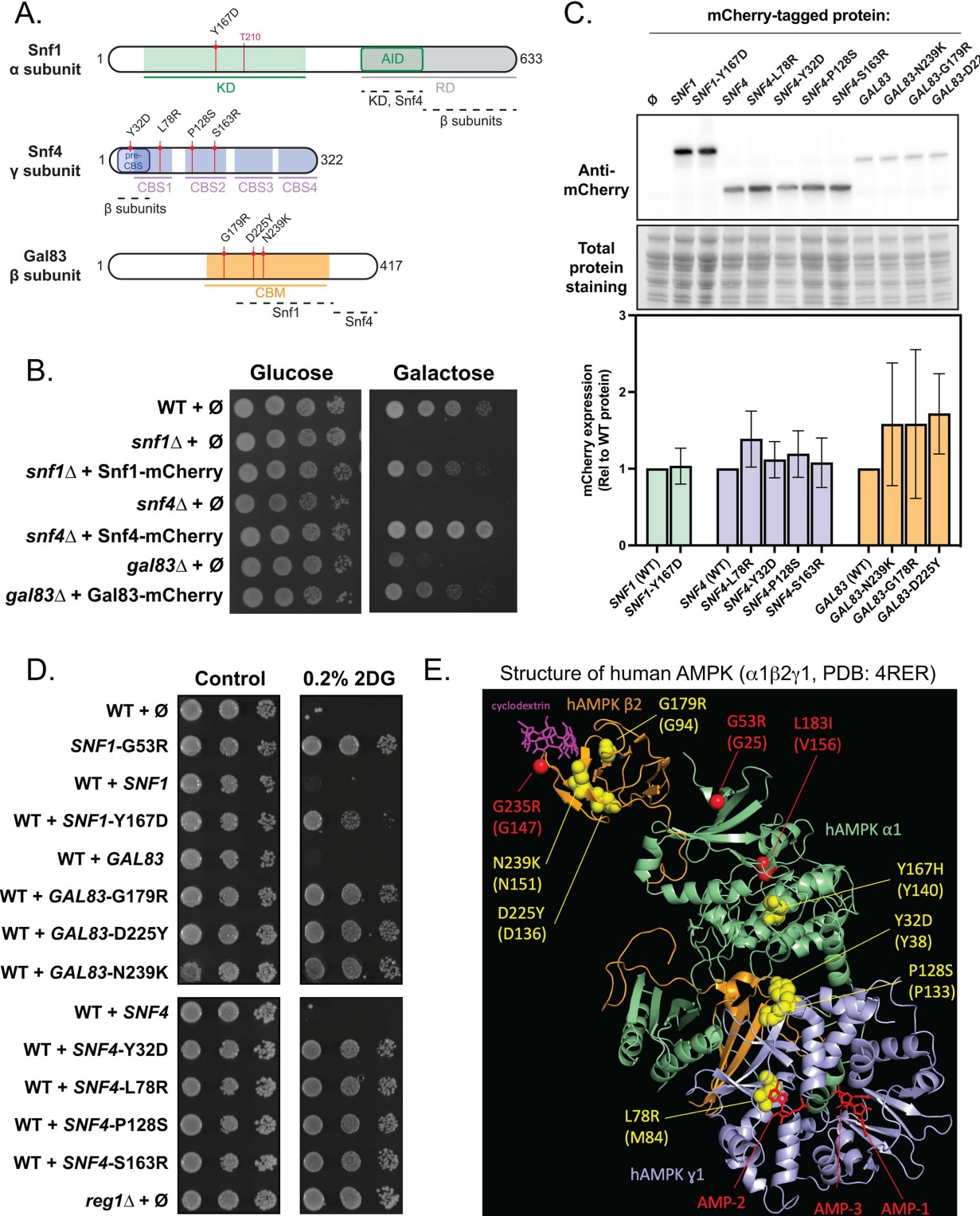

**Fig 2. Dominant alleles in three subunits of SNF1/yAMPK. A.** Schematic of the three proteins composing the SNF1 complex, highlighting their domains and the position of the mutations found in the genetic screen. KD: kinase domain; AID: auto-inhibitory domain; CBM: carbohydrate-binding module; CBS: cystathionine beta-synthase domains. **B.** Growth assay of the mentioned strains, transformed with plasmids expressing the indicated

mCherry-tagged proteins. Serial dilutions of yeast cultures were spotted on a synthetic medium (uracil drop-out) containing 2% glucose or 2% galactose and grown for 3 days at 30°C. **C.** *Top*, Total protein extracts of cells from the mentioned genotype, transformed with plasmids expressing the indicated WT or mutant protein, tagged with mCherry, were immunoblotted using an anti-RED antibody. *Bottom*, Histogram showing the abundance of the indicated mutant proteins, normalized to total proteins and relative to the corresponding WT protein ($n = 4$; ± SEM). Paired *t*-test revealed no significant expression changes between any of the mutants and their corresponding WT protein. **D.** Growth assay of the mentioned strains, transformed with plasmid expressing the indicated mCherry-tagged protein. Serial dilutions of yeast cultures were spotted on synthetic medium (uracil drop-out) containing 2% glucose supplemented or not with 0.2% 2DG, and grown for 3 days at 30°C. **E.** Position of the residues targeted by mutations on the structure of human AMPK (α1β2γ1, PDB: 4RER [68]). Orange: hAMPK β2 subunit, green: hAMPK α1 subunit, blue: hAMPK γ1 subunit. Residues targeted by mutations and conserved in human proteins are shown in yellow (this study) or in red (previous studies, see text). The position of the conserved residue in the human protein is indicated between parentheses. Ligands such as AMP molecules and β-cyclodextrins are indicated in red and purple, respectively.

sufficient to confer 2DG resistance (Fig 2D), similarly to the previously published gain-of-function mutation *SNF1-G53R* [46,55,69,70]. Thus, these mutants are sufficient to cause 2DG resistance and act in a dominant way.

The mutation identified in *SNF1*, causing a Y167D substitution in the kinase domain, is analogous to the Y167H mutation previously found to render Snf1 activity constitutive and independent of Snf4 [57]. Two mutations in *GAL83* targeted the carbohydrate-binding module (CBM), causing either D225Y or N239K substitutions. The D225Y mutation was previously isolated during evolution experiments and found to improve the use of xylose as a carbon source [71], and is nearby the S224R mutation previously isolated as causative of 2DG resistance [55]. Moreover, genome resequencing of a dominant mutant we previously isolated (clone #23 from [53]) revealed another mutation in the CBM domain (G179R). Mutation or deletion of the CBM domain of Gal83 was previously found to increase Snf1 activity and lead to 2DG resistance [55,58,72]. Finally, we also obtained 4 independent mutations in *SNF4*, causing substitutions in the Bateman 1 domain (CBS1 and CBS2): Y32D, L78R, P128S, and S163R. Based on the published structure of the human AMPK complex (PDB accession: 4RER [68]; Fig 2E), Y32 and P128 are located nearby in a region mediating interaction with the beta-subunit [73]. The L78R mutation, instead, targets a residue (M84 in hAMPK β2) facing AMP in the AMP-2 binding site [73]. Whereas prior studies showed that mutations in this region of Snf4 can lead to a constitutive Snf1 activity and 2DG resistance [55,59], none of the mutations we obtained were previously identified and they may represent novel gain-of-function alleles.

## Analysis of mutants in PP1-encoding genes (*GLC7* and *REG1*) reveals two functional classes

PP1, composed of its catalytic subunit Glc7 and the regulatory subunit Reg1, is a well-known negative regulator of Snf1 activity. Glc7 serves as a common catalytic subunit for all yeast PP1 complexes, consequently this gene is essential for viability [for review, see 74]. Previous studies isolated viable mutants in *GLC7* [75], including mutation Q48K found to cause 2DG resistance and increased SNF1 signaling [46]. Whole-genome resequencing of a mutant we previously isolated (clone #5) [53] identified the analogous mutation Q48P.

We isolated three new viable mutations in *GLC7* (causing substitutions N85D, Y254C and Q293P; Fig 3A) from 5 different clones (S1 and S2 Tables). Their growth in presence of 2DG was confirmed, and we also evaluated their resistance to drugs which are toxic to *reg1Δ* strains (Fig 3B). This included tunicamycin, a N-glycosylation inhibitor whose sensitivity was previously linked to PP1/AMPK signaling [77,78], and selenite, which enters through the Jen1 lactate transporter whose expression is upregulated in the *reg1Δ* strain [79,80]. The rationale was that mutations in *GLC7* causing 2DG resistance might mimic a loss of *REG1* function. Iodine treatment was also performed to evaluate the glycogen content of the isolated mutants, of which Glc7 is a known regulator [75], and because *reg1Δ* also accumulates glycogen [39]. Overall, the isolated *glc7* mutants did not recapitulate the *reg1Δ* phenotype, showing no sensitivity to selenite nor tunicamycin, and variable glycogen content (Fig 3B). Yet, 2DG resistance was abolished upon expression of a WT copy of *GLC7* tagged with GFP, suggesting that the identified mutations are responsible for this phenotype (Fig 3C). Previous work has shown that mutations in *GLC7* often affect the binding to multiple regulatory subunits and thus the function of multiple PP1

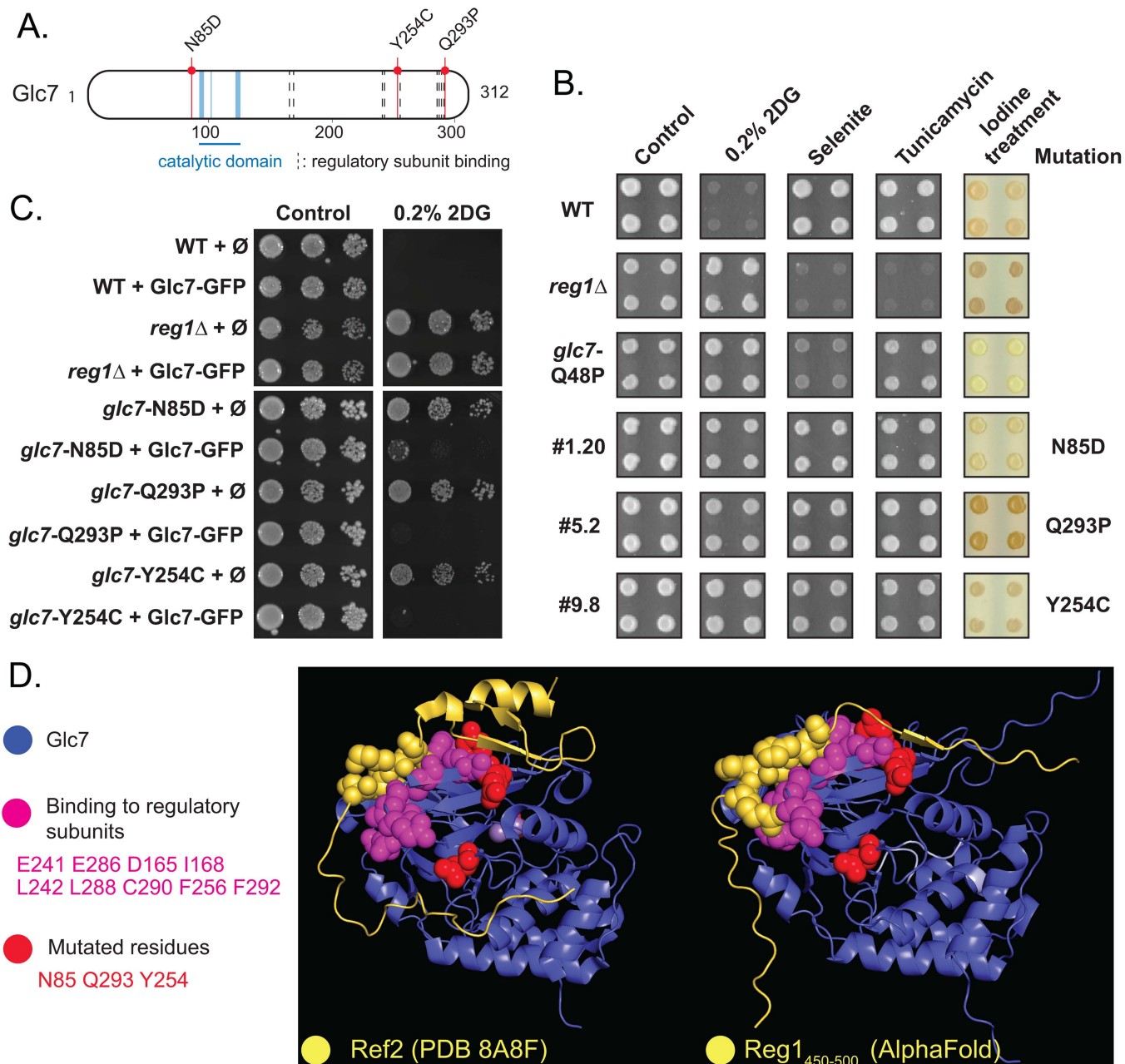

**Fig 3. Analysis of mutants in *GLC7*. A.** Schematic representation of the Glc7 protein. The catalytic domain, residues involved in the binding of regulatory subunits and mutations found in the screen are highlighted. **B.** Phenotypes of the indicated strains (WT, *reg1Δ*, control *glc7*-Q48P and three clones from the screen carrying a mutation in *GLC7*) on plate. Saturated liquid cultures were pinned in quadruplicate on the indicated media using a robot for high-throughput screening and incubated at 30°C for 3 days. Iodine treatment was performed by exposing the plates to iodine crystals for 2 minutes before scanning. **C.** Growth assay of the indicated strains complemented or not with a centromeric plasmid allowing the expression of Glc7-GFP under the control of its endogenous promoter. Serial dilutions of yeast cultures were spotted on synthetic medium (uracil drop-out) containing 2% glucose, supplemented or not with 0.2% 2DG, and grown for 3 days at 30°C. **D.** Position of the residues targeted by mutations on the structure of Glc7. *Left,* three-dimensional structure of Glc7 in complex with a portion of its regulatory subunit Ref2 (PDB accession: 8A86; [76]); *Right:* AlphaFold modeling of Glc7 in complex with a portion of Reg1 containing the RHIHF motif required for interaction with Glc7 (residues 450-500). Blue: Glc7; Yellow: Ref2/Reg1, spheres indicate residues composing the degenerate RVxF motif; Pink: residues of Glc7 known to be involved in the binding to its regulatory subunits; Red: residues for which mutations were identified in this study.

complexes [75], complicating the study of these mutants which display pleiotropic phenotypes. Indeed, several residues that we found mutated in Glc7 (e.g., Y254 and Q293) are nearby residues demonstrated to be involved in binding to regulatory subunits through their canonical RVxF motif (which include F256 and F292 on Glc7) [81,82]. Mapping of these residues on the published structure of Glc7 in complex with the regulatory subunit Ref2 (PDB 8A8F) [76] (Fig 3D) suggests a possible general interference with binding to regulatory subunits, and so does the AlphaFold prediction of the Glc7/Reg1 interaction (Fig 3D). Residue N85, however, was further away from the canonical regulatory subunit-binding site and may be involved in binding to additional motifs carried by regulatory subunits [76,83]. Noteworthy, mutations in this region of Glc7 were previously isolated (e.g., P82S) causing defective binding to various subunits [75]. Altogether, we conclude that whereas the newly isolated mutations in Glc7 are responsible for the 2DG resistance phenotype, they are likely to be pleiotropic, which may complicate our study of AMPK regulation by glucose availability.

Consequently, we focused our analysis on mutations in *REG1*. Many mutants (72) were ascribed to the *REG1* complementation group. Interestingly, quantification of growth of these mutants even in the absence of 2DG revealed two classes within this group. Approximately half (37) of the obtained diploids displayed a slow growth in the control medium, as expected for *reg1/reg1*Δ mutants (red squares, class I), whereas another class (35 mutants, class II) showed a faster growth (blue squares) (Fig 4A and S2 Table). To further confirm that these mutants are *reg1* mutants, the original haploid mutants were spotted on 2DG-, selenite- or tunicamycin-containing medium. Class I mutants displayed similar sensitivity profiles as the *reg1*Δ mutant, whereas class II were insensitive to these drugs (Fig 4A). They also appeared to accumulate less glycogen than *reg1*Δ.

We sequenced the *REG1* gene in 72 mutants and found that the class I mutants indeed carried a nonsense mutation within the first 725 residues of the protein (Fig 4B And S1 and S2 Tables). These mutants were of limited interest as they are functionally equivalent to *REG1* deletion, for which 2DG resistance is already well documented [45,46,52,53]. Other mutants in this class carried mutations in the *REG1* promoter (clones 5.9 and 7.18), or frameshift mutation (4.4 and 7.3), internal duplication (10.12), missense mutations at the start codon (7.2, 10.4) or at the Glc7-binding site (F468C, clone 10.5). Instead, mutants of class II all carried a missense mutation in conserved residues, located in conserved regions of the proteins (Fig 4C and S1 and S2 Tables). Two additional mutants of this class were identified by whole-genome resequencing of previously isolated mutants [53], carrying point mutations D146Y (clone #18) and P231S (clone #22). The latter is analogous to the P231L mutation previously described as causing 2DG resistance [46]. Moreover, one mutant obtained from a preliminary screen carried two point-mutations in the Glc7 binding site (RHIHF to RYIYF). Altogether, we report 26 novel mutations in *REG1* that confer 2DG resistance without causing other phenotypes typically displayed by the *reg1*Δ mutant. This suggests that these are separation-of-function alleles that bypass 2DG toxicity without fully recapitulating the effects of a complete *REG1* deletion.

## Missense mutations in *REG1* confer 2DG resistance but no apparent glucose-repression defects

We followed up on the missense mutations by confirming the role of the missense mutations in the resistance phenotype. We selected all 17 *REG1* mutants that were available at the initiation of this work, except when two mutations targeted the same residues (E195, P278) in which case we picked only one mutant. We did not study mutants which, based on the phenotypes, appeared as loss-of-function mutations – this included the F468C mutant which carried a mutation in the RHIH**F** motif for interaction with Glc7 [84], and point mutations in the start codon (M1R, M1I) which are likely to prevent translation. The selected mutants were cloned into centromeric plasmids to allow expression of the mutant proteins under the control of the endogenous *REG1* promoter with a FLAG tag, which did not impact on the WT protein's function (S2 Fig). These mutants were introduced in a *reg1*Δ strain, and the resulting strains were spotted in quadruplicate onto synthetic medium containing 2DG, selenite or no drug. Growth was quantified by measuring colony size after 3 days of growth, and the results are shown as a heat map with growth normalized to the control (Fig 5A). In agreement with our conclusions obtained from our original strains isolated from the screen (see Fig 4A), we found that expression of these mutants suppressed the slow growth phenotype of the *reg1*Δ strain, as well as its tunicamycin- and selenite-sensitivity phenotypes, but not its resistance to 2DG.

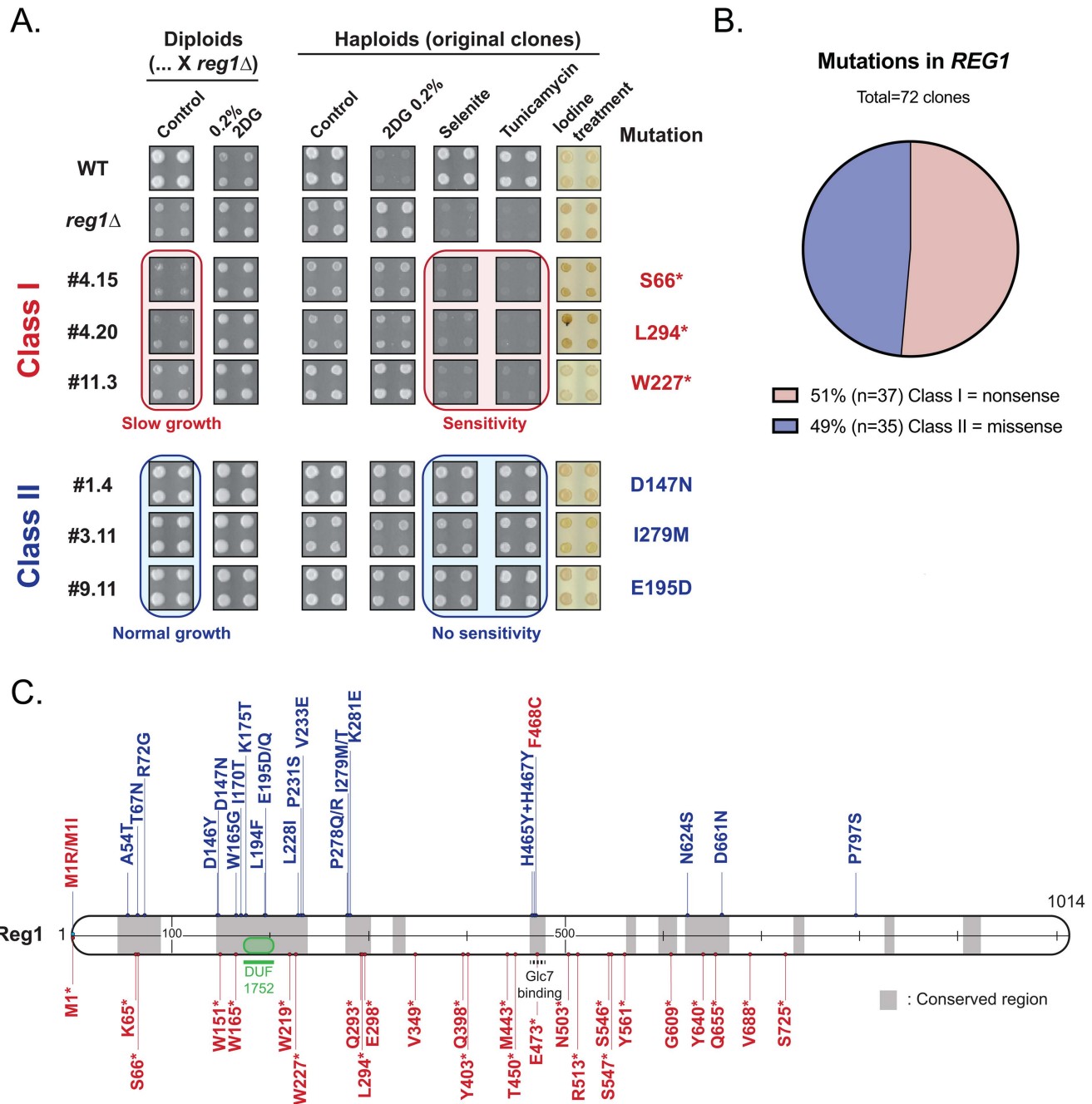

**Fig 4. Analysis of mutants in *REG1*. A.** Phenotypes of the indicated strains (haploids or diploids, as indicated) on plate. Saturated liquid cultures were pinned in quadruplicate on the indicated media using a robot for high-throughput screening and incubated at 30°C for 3 days. Iodine treatment was performed by exposing the plates to iodine crystals for 2 minutes before scanning. The growth of a few *REG1* mutants and of control strains is shown. Representatives of class I mutants (red) and class II mutants (blue) are indicated. **B.** Pie-chart showing the proportion of nonsense (class I) and missense (class II) mutations predicted from phenotypic characterization of mutants assigned to the *REG1* complementation group (see S2 Table). **C.** Schematic representation of the Reg1 protein. A putative domain of unknown function (DUF1752) is indicated, together with class I (red) and class II (blue) mutations identified in the genetic screen. M1R and M1I are point mutations in the start codon causing substitution into Arg and Ile, respectively, and behave like a nonsense mutation. Mutation F468C targets the Glc7-interaction motif (RHIHF) and also behaves like a nonsense mutation. Asterisks indicate the position of the stop codon in the protein.

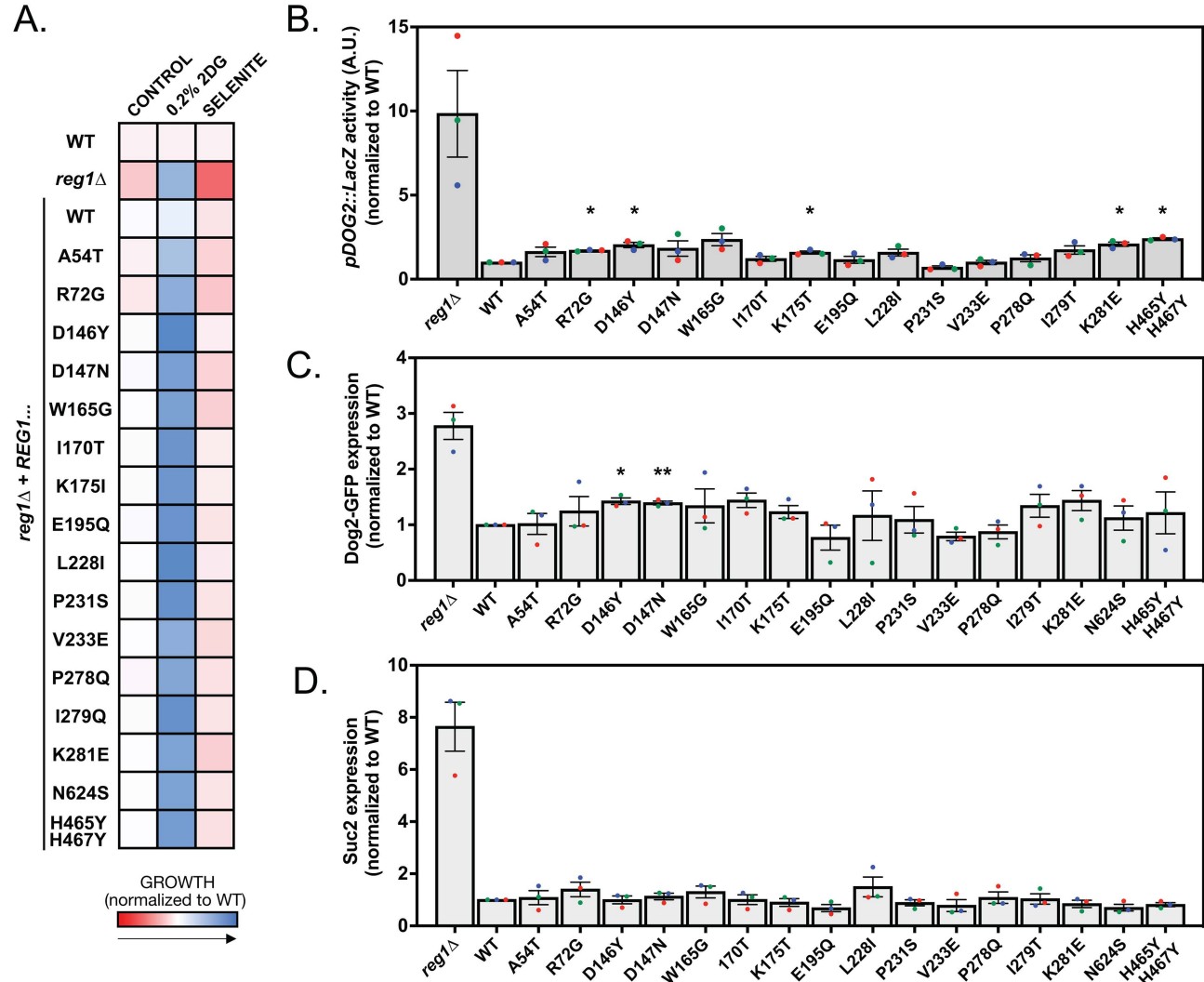

**Fig 5. Mutants carrying missense mutations in Reg1 are competent for glucose repression. A.** Heatmap displaying the normalized growth of WT, *reg1Δ* or *reg1Δ* cells transformed with a plasmid expressing the indicated Reg1-FLAG allele. Saturated liquid cultures were pinned in quadruplicate on the indicated media using a robot for high-throughput screening and incubated at 30°C for 3 days before scanning. Growth was normalized to the WT in each condition. **B.** Histogram representing the activity of a *pDOG2::lacZ* construct in *reg1Δ* cells transformed with plasmid expressing the indicated Reg1-FLAG allele, normalized to WT signal ($n = 3, \pm$ SEM). **C, D.** Histogram representing the normalized (to WT) abundance of Dog2-GFP (**C**) and Suc2 (**D**) in *reg1Δ* cells transformed with plasmids expressing the indicated Reg1-FLAG allele. Quantification of western blots on total protein extracts of cells growing in exponential phase after overnight growth at 30°C. ($n = 3 \pm$ SEM). For panels B-D, * and ** indicate the *p*-value of statistical comparisons with respect to the strain expressing WT Reg1-FLAG. *: $p < 0.05$; **: $p < 0.01$. No indication means not significant.

The observed lack of selenite sensitivity of these mutants suggested a normal expression of the selenite transporter Jen1, which is repressed in the presence of glucose but aberrantly derepressed in the *reg1Δ* strain [79,80]. Similarly, the lack of tunicamycin sensitivity, which is linked to a constitutively high AMPK activity in the *reg1Δ* mutant [78,85], also indicated that AMPK signaling is not upregulated in this mutant. To confirm this hypothesis, we looked at the expression of a LacZ reporter fused to the *DOG2* promoter, that we previously showed to be in part regulated by Snf1 through a Mig1-regulated repression [53]. Whereas we could detect a strong expression of the reporter in the *reg1Δ* strain, none of

the isolated mutants showed a strong increase in *pDOG2-LacZ* expression (Fig 5B). This was confirmed by monitoring the expression of proteins encoded by glucose-repressed genes by immunoblotting. A *reg1Δ* strain expressing *DOG2-GFP* tagged at its endogenous locus was transformed with the various *REG1* alleles and Dog2-GFP expression was quantified (Fig 5C). This was confirmed by immunodetection of invertase, encoded by the glucose-repressed gene *SUC2* [86], showing that these point mutants maintained the glucose-repression of this gene (Fig 5D). Thus, apart from their 2DG resistance, the *REG1* missense point mutants behave similarly to the WT and display a normal inhibition of AMPK pathway activity.

## Reg1 missense mutations vary in their ability to dephosphorylate Snf1 and its substrate Mig1 in response to 2DG

We focused on five mutants for a further functional characterization (Fig 6A). We selected mutations located in several of conserved regions, and included the I466M/F468A (also known as "IMFA") mutant which lacks the ability to interact with Glc7 [84]. The expression level was not drastically affected by the mutations (Fig 6B), except for the IMFA mutant which consistently showed a twofold reduction in protein levels, in agreement with previous observations [84].

A frequent consequence of mutations is a modification of protein-protein interaction. Reg1 is known to interact with Glc7 [37,84], with 14-3-3 proteins [87,88] and with components of the AMPK complex including Snf1 [36]. We thus checked the ability of the Reg1 mutants to interact with these proteins by co-immunoprecipitation (Fig 6C and 6D). This revealed no significant changes in interactions at steady-state.

We then studied the phosphorylation of Snf1 and its target Mig1 (Fig 7A-C). In glucose-grown conditions, the phosphorylation of these proteins was similar in cells expressing WT or mutant alleles of Reg1, in agreement with our hypothesis that Snf1 signaling is not affected (no growth defect, no aberrant *DOG2* or *SUC2* expression: Figs 4 and 5). We then studied the response of the mutants to 2DG treatment, having previously shown that 2DG treatment leads to Snf1 and Mig1 dephosphorylation [50] (Fig 7A-C). This revealed a heterogeneous response among the various mutants. One mutant, W165G, failed to dephosphorylate both Snf1 and Mig1 in response to 2DG, and therefore seemed 2DG-insensitive. Two other mutants, P231S and P278Q, showed a defective dephosphorylation of Snf1 but only a partial dephosphorylation of Mig1. Finally, the ability of the A54T and N624S mutants to dephosphorylate Snf1 and Mig1 was comparable to that of the WT. Therefore, the five mutants we selected seem to be differentially affected for PP1 activity towards Snf1 or its substrate Mig1 in response to 2DG, with mutant W165G being the most affected mutant.

Since 2DG toxicity depends on its phosphorylation, we then assayed 2DG6P content of these mutants shortly after 2DG treatment to see whether it correlates with PP1 activity towards the examined substrates (Fig 7D). First, we observed that the *reg1Δ* mutant does not contain 2DG6P within 15 min of 2DG treatment, which may contribute to its strong resistance to 2DG and its apparent insensitivity to this drug, although we did not address whether 2DG6P would accumulate at later timepoints. In contrast, the *REG1* mutants tested (W165G, A54T, P231S) accumulated 2DG6P to a level that was not significantly different from the WT, suggesting that the lack of 2DG response was not due to an absence of 2DG phosphorylation.

These results were confirmed by measuring the effect of 2DG on ATP levels using an optimized, pH-insensitive version of the AT1.03 ATP FRET biosensor [89]. The phosphorylation of 2DG into 2DG6P is known to be accompanied by a decrease in ATP content, which we observed in *reg1Δ* cells complemented with WT Reg1 but also with the REG1 mutants tested (Fig 7E). In contrast, ATP levels were less affected in the *hxk2Δ* mutant, which lacks the main isoform of hexokinase in charge of 2DG phosphorylation [50], and even less so in the *reg1Δ* strain. Altogether, these data demonstrate that the differential behavior of the studied *REG1* mutants did not seem to correlate with 2DG6P phosphorylation, but rather revealed a relative inability of these mutants to relay the 2DG signal. They also reveal a new strategy of resistance to 2DG, which does not involve a constitutively high AMPK activity or a decreased 2DG6P level, but rather a lack of inhibition on the AMPK pathway in response to 2DG.

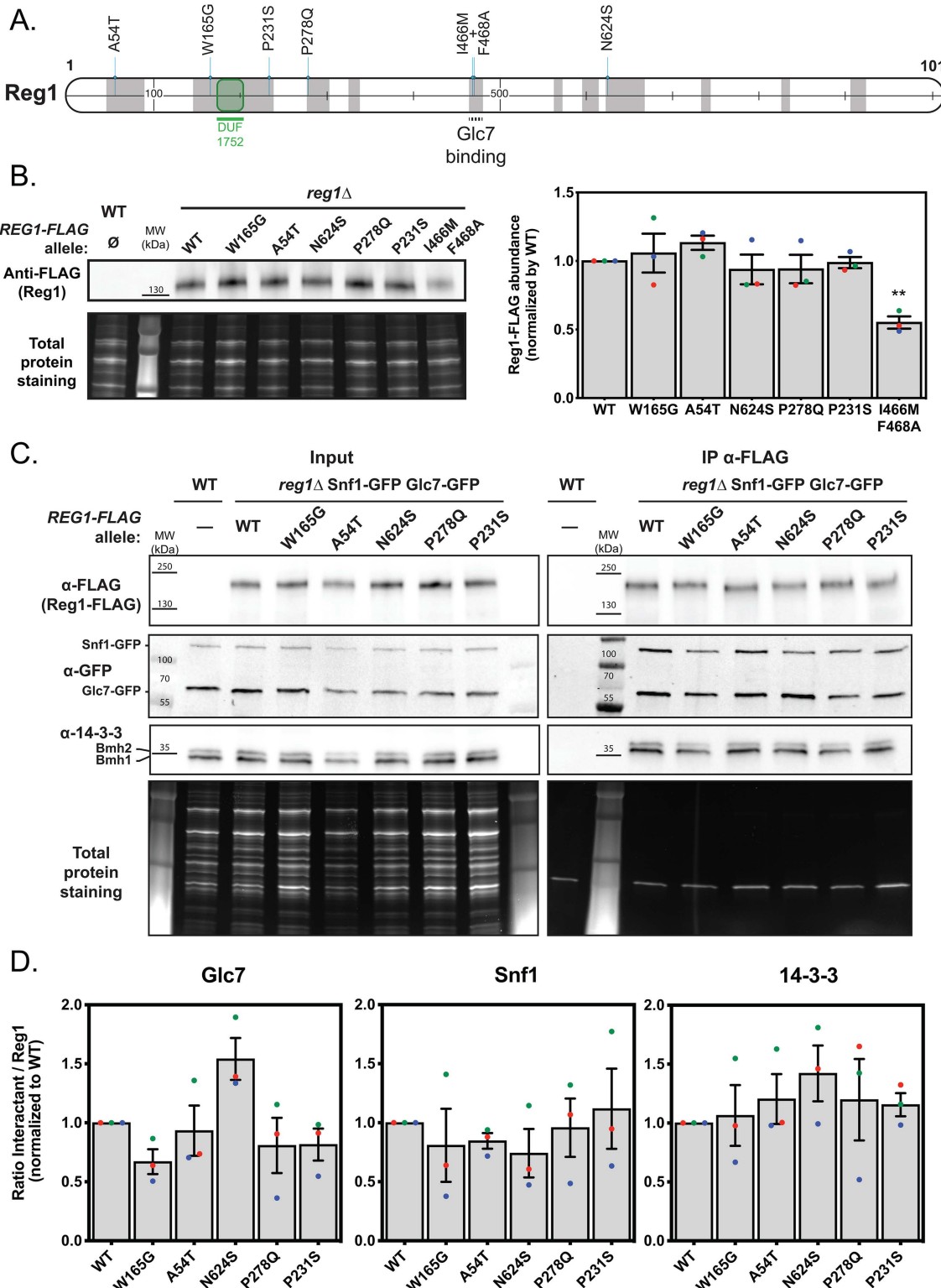

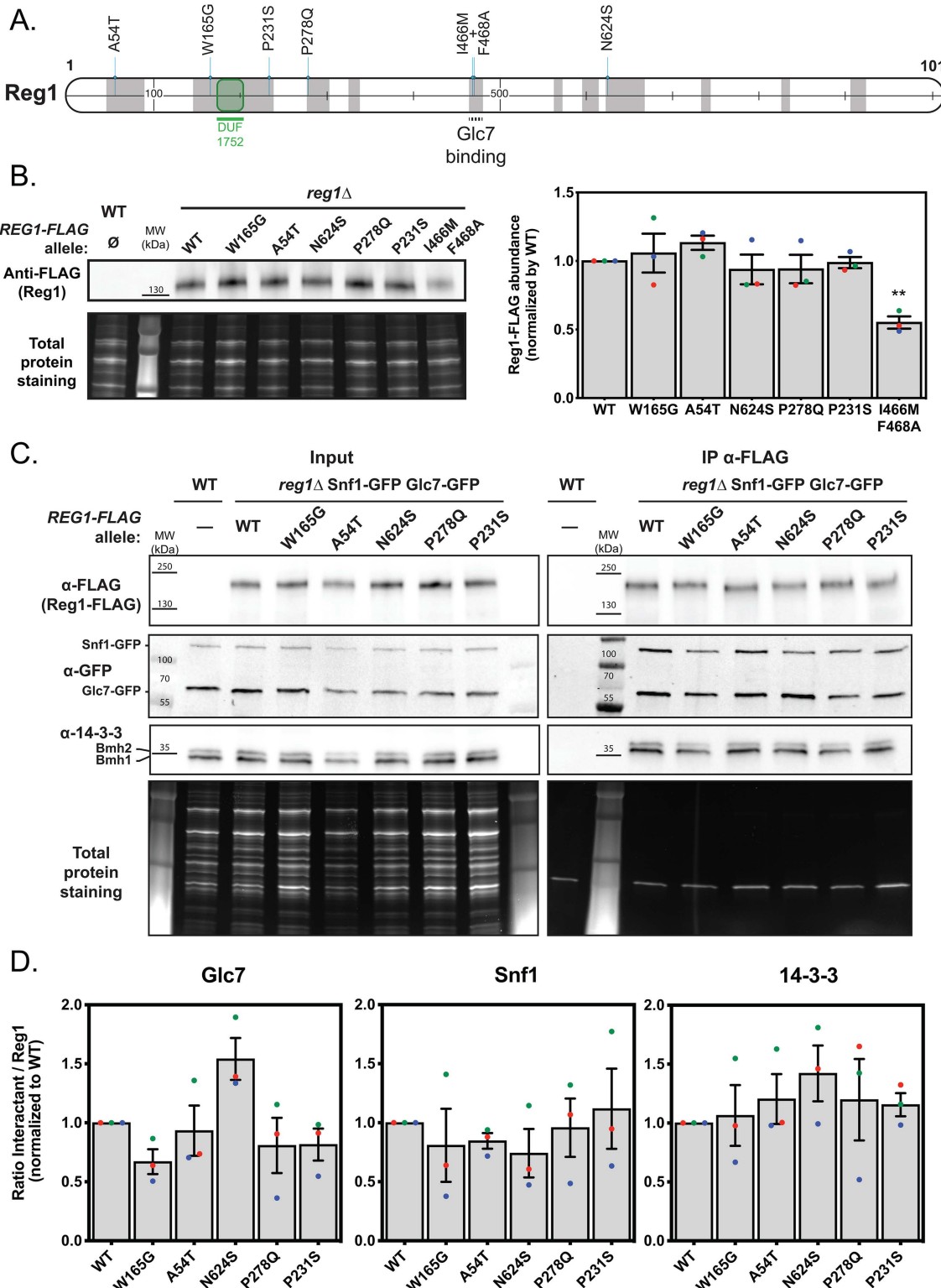

**Fig 6. Initial characterization of Reg1 point mutants. A.** Schematic representation of the Reg1 protein. The DUF1752 domain is highlighted, together with the five missense mutations carried by the mutants further characterized in this study. **B.** *Left*, Total protein extracts of cells from the mentioned genotype, transformed or not with plasmids expressing the indicated Reg1-FLAG alleles, were immunoblotted using an anti-FLAG antibody. *Right,*

Histogram showing Reg1, WT or mutant, abundance normalized to the WT. **C.** Total protein extracts of cells from the mentioned genotype, transformed or not with plasmids expressing the indicated Reg1-FLAG alleles, were immunoprecipitated using anti-FLAG antibodies and immunoblotted using anti-FLAG (Reg1), anti-GFP (Glc7 and Snf1) and anti-14-3-3s (14-3-3 proteins) antibodies. *Left*, total extracts before immunoprecipitation, *Right*, immunoprecipitations. **D.** Histogram displaying the quantification of the ratio of the detected signal for each protein over Reg1-FLAG, detected in the immunoblots of **B. B and D.** $n = 3$. error bars = SEM, ** indicate the *p-value* of statistical comparisons: **: $p < 0.01$. No indication means not significant.

### Missense mutations in *REG1* impact on the dephosphorylation of the arrestin-related protein Rod1 and 2DG-induced endocytosis of the glucose transporter, Hxt3

We further documented the signaling defects displayed by the various Reg1 mutants in response to 2DG by looking at the dephosphorylation of the arrestin-related protein Rod1, which we also found to be rapidly dephosphorylated in response to 2DG [50]. This dephosphorylation event is followed by the ubiquitylation of a fraction of the protein, resulting in a doublet by SDS-PAGE [50,56,60,61]. After 2DG treatment, a doublet was indeed observed for the *reg1*Δ strain complemented with WT Reg1 (Fig 8A). However, mutant W165G failed to dephosphorylate Rod1 in response to 2DG, and again seemed unresponsive to 2DG. Two other mutants, P231S and P278Q, showed only a partial dephosphorylation of Rod1. Finally, both the A54T and N624S mutants were able to dephosphorylate Rod1 to a similar extent as WT Reg1. Thus, the response of Rod1 parallels that of Mig1 and is consistent with the results obtained for Mig1 (Fig 7A). Noteworthy, the analysis of Snf1 phosphorylation in the course of these experiments (Fig 8B) matched the results obtained previously (Fig 7A and 7C). This revealed again the heterogeneity of these mutants regarding their ability to dephosphorylate Snf1 and its targets.

There is a body of evidence showing that Rod1 dephosphorylation and subsequent ubiquitylation promotes its activity in endocytosis [50,56,60,61]. The overexpression of several hexose transporters was reported to confer 2DG resistance [56] and we previously proposed that the lack of hexose transporter endocytosis results in 2DG resistance [50]. Thus, we examined the ability of a few Reg1 mutants to internalize the glucose transporter Hxt3, which is known to be endocytosed in response to 2DG in a Rod1-regulated manner [50,56]. Some level of endocytosis was detected in all mutants upon treatment with 2DG (Fig 8C), but the quantifications (Fig 8D) showed differences in the response. Whereas endocytosis in Reg1-A54T and Reg1-P231S was not very different from that observed in WT cells, it was strongly affected in the Reg1-W165G mutant. This was in agreement with our data showing the lack of a 2DG-induced change in Rod1 post-translational modifications in the latter strain (Fig 8A). Thus, it is possible that part of the 2DG resistance of the Reg1-W165G mutant is due to a decreased endocytosis of Hxt3, as shown before for *rod1*Δ [50].

In the course of these experiments, we also confirmed that 2DG induces vacuolar fragmentation [50,90]. This was also observed in cells expressing the Reg1-A54T mutant, but not as much for the Reg1-P231S mutant and fragmentation did not seem to occur for the Reg1-W165G mutant (Fig 8C). The cause of this fragmentation is uncertain, but ER stress triggers this phenotype [91], and 2DG causes ER stress by interfering with protein N-glycosylation [53]. However, it reveals once again a heterogeneity among the Reg1 mutants we obtained in their response to 2DG.

### Mutation W165G in Reg1 renders Snf1 regulation specifically insensitive to 2DG

The Reg1 mutant W165G was of peculiar interest because of its strong lack of response to 2DG regarding the dephosphorylation of Snf1, Rod1 and Mig1. We hypothesized that in response to 2DG, this mutant is unable to behave like WT Reg1 and that this may involve differential interactions upon 2DG treatment. Because we could not see any difference in interactions with known partners at steady-state between WT Reg1 and the W165G mutant (see Fig 6C-D) we first sought to establish differential interactants of WT Reg1 before and after 2DG treatment by quantitative interaction proteomics (Fig 9A and S3 Table). A strain expressing Reg1-FLAG was subjected to FLAG immunoprecipitation before and after (10 min) treatment with 2DG. Overall, changes in interactants were minor, and we could notably observe that Reg1 interaction with Glc7 or 14-3-3 did not change with 2DG treatment (Fig 9A and quantifications in Fig 9B). However, subunits of AMPK as well as Rod1 seemed enriched in the IP after 2DG treatment (Fig 9A and 9B). In contrast, when this experiment was

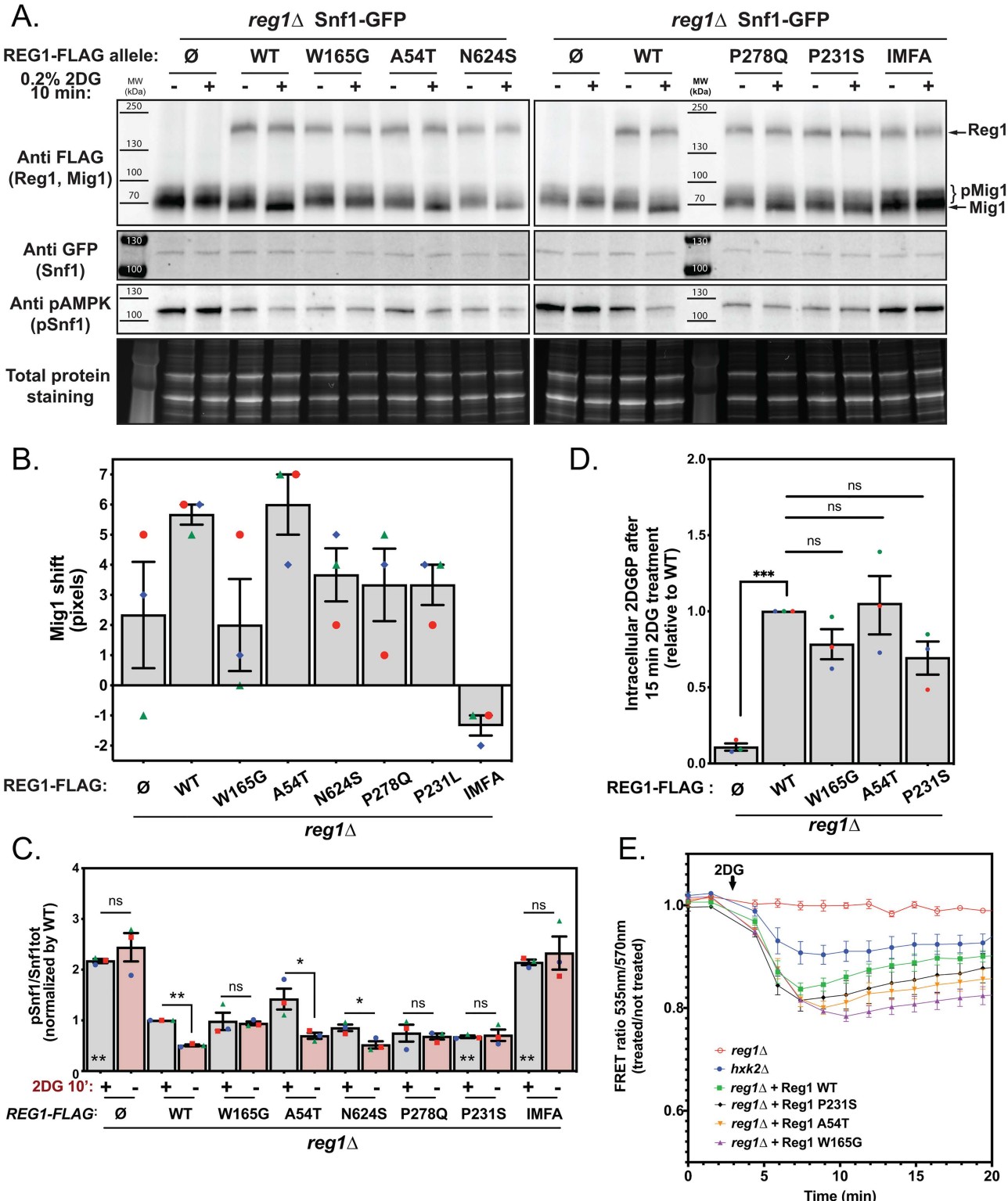

**Fig 7. Characterization of the 2DG response in Reg1 point mutants. A.** Total protein extracts of cells from the mentioned genotype, expressing Snf1-GFP (endogenously tagged) and Mig1-FLAG (centromeric plasmid) and transformed or not with a plasmid expressing the indicated Reg1-FLAG alleles, were prepared before and after 2DG addition for 10', and immunoblotted using anti-FLAG (Reg1 and Mig1), anti-GFP (Snf1) and anti-pAMPK (pSnf1)

antibodies. **B.** Histogram representing the shift of Mig1 (0' vs. 10' treatment) in the indicated mutant in pixels, as measured using the ImageQuant TL software. **C.** Histogram representing Snf1 phosphorylation state (pSnf1/Snf1tot) in the mentioned strains, before or after 2DG treatment. * on top of the bars: 0' vs. 10' 2DG treatment, * inside the bars: comparison of the 0' 2DG time point to the WT. **D.** Intracellular 2DG6P was assayed enzymatically (see Methods) in *reg1*Δ, WT and *reg1*Δ cells transformed with the indicated Reg1-FLAG alleles. Cells were grown overnight in a glucose-containing medium and treated or not for 15 min with 0.2% 2DG. Values are normalized to the value of the WT (15'). **E.** Cells expressing an ATP FRET biosensor and the mentioned mutant Reg1-FLAG protein were grown overnight in a glucose-containing medium and treated with 0.2% 2DG (arrow). The FRET ratio (535/570 nm) was measured over time in a plate reader (see Methods) and is represented as the ratio between treated and non-treated cells ($n = 4$). **B-D**. $n = 3$, error bars = SEM, * indicate the *p-value* of statistical comparisons: *$P < 0.05$; **$P < 0.01$.

performed using the Reg1-W165G mutant, we saw no enrichment of AMPK subunits nor of Rod1 after 2DG treatment (Fig 9C and quantifications in Fig 9D). This indicates that this point mutation prevents the 2DG-induced binding of PP1 to the AMPK complex and to Rod1, in line with the observed absence of dephosphorylation.

The possibility that mutation W165G does not cause a general loss of PP1 function is supported by the fact that Snf1 is not constitutively activated when cells are grown in glucose (see Figs 6-8). Thus, it seems that this allele is specifically affected in its reaction to 2DG, in line with the design of the screen that selected 2DG-resistant mutants in the presence of glucose, and their further phenotypic selection as alleles that do not cause constitutive Snf1 signaling. In agreement with this, 2DG did not prevent the growth of a strain expressing the Reg1-W165G mutant in sucrose medium, suggesting that *SUC2* expression is also 2DG-insensitive (Fig 9E). Instead, we found that glucose-starved cells expressing the Reg1-W165G mutant were able to dephosphorylate Snf1, Mig1 and Rod1 in response to glucose, albeit with a reduced efficiency (Fig 9F-G). Thus, we propose that the W165G mutation causes a failure in specifically relaying the 2DG signal, likely through a lack of induced recruitment to its substrates.

## Discussion

In this study, we aimed at better understanding the molecular basis of glucose signaling and 2-deoxyglucose resistance by identifying mutants growing on 2DG in the presence of glucose as a carbon source. This allowed us to identify mutations in many genes, most notably involved in the Snf1/AMPK pathway or its regulators, which confirm and extend previous findings [46,52,53,55]. Through our detailed analysis of several point mutants, in particular Reg1-W165G, our data support a model in which Reg1 may act as a sensor of metabolic stress to regulate SNF1 activity.

Among the most represented genes identified in the genetic screen are *HXK2*, the main isoform of hexokinase in glucose-grown cells. Hxk2 acts both as a hexose phosphorylating enzyme and glucose signaling protein, although it is still unclear whether these functions are linked, and both contribute to 2DG resistance. Whereas Hxk1, Hxk2 and Glk1 are all able to phosphorylate 2DG [46], only *HXK2* deletion causes 2DG resistance [45,46,52,53] and we reported that Hxk2 is the main 2DG-phosphorylating enzyme *in vivo* [50]. Consequently, deletion of *HXK2* limits the generation of 2DG6P in cells, leading to a lower toxicity. Moreover, the absence of Hxk2 is associated with an increased Snf1 phosphorylation [45,46] despite the presence of other hexose/glucose kinases (Hxk1, Glk1), and increased Snf1 activity is a hallmark of many isolated 2DG-resistant mutants [46,52,53,55] (see also discussion below).

In line with the fact that Snf1 hyperactivation leads to 2DG resistance, many mutants were associated with a complete loss of function of *REG1* (nonsense mutation) or with missense mutations in *GLC7*. In addition, several new mutations in *SNF1, SNF4* or *GAL83* were found as dominant alleles which were sufficient to confer 2DG resistance to WT cells, suggesting they are gain-of-function mutations, as previously identified in a screen [55]. Again, Snf1 hyperactivation is likely to cause 2DG resistance by several mechanisms. First, we previously reported that the expression of a small metabolite phosphatase conferring 2DG resistance, named Dog2 [92,93], is positively regulated by Snf1 [53]. Dog2 expression is induced upon 2DG exposure but is also repressed in presence of glucose through Mig1, causing a higher expression in glucose-repression mutants and thus possibly counteracting the accumulation of 2DG6P [53]. Second, Snf1 is known to inhibit the function of the arrestin-related protein Rod1 in endocytosis [56,60,61], and we found that 2DG-induced endocytosis of

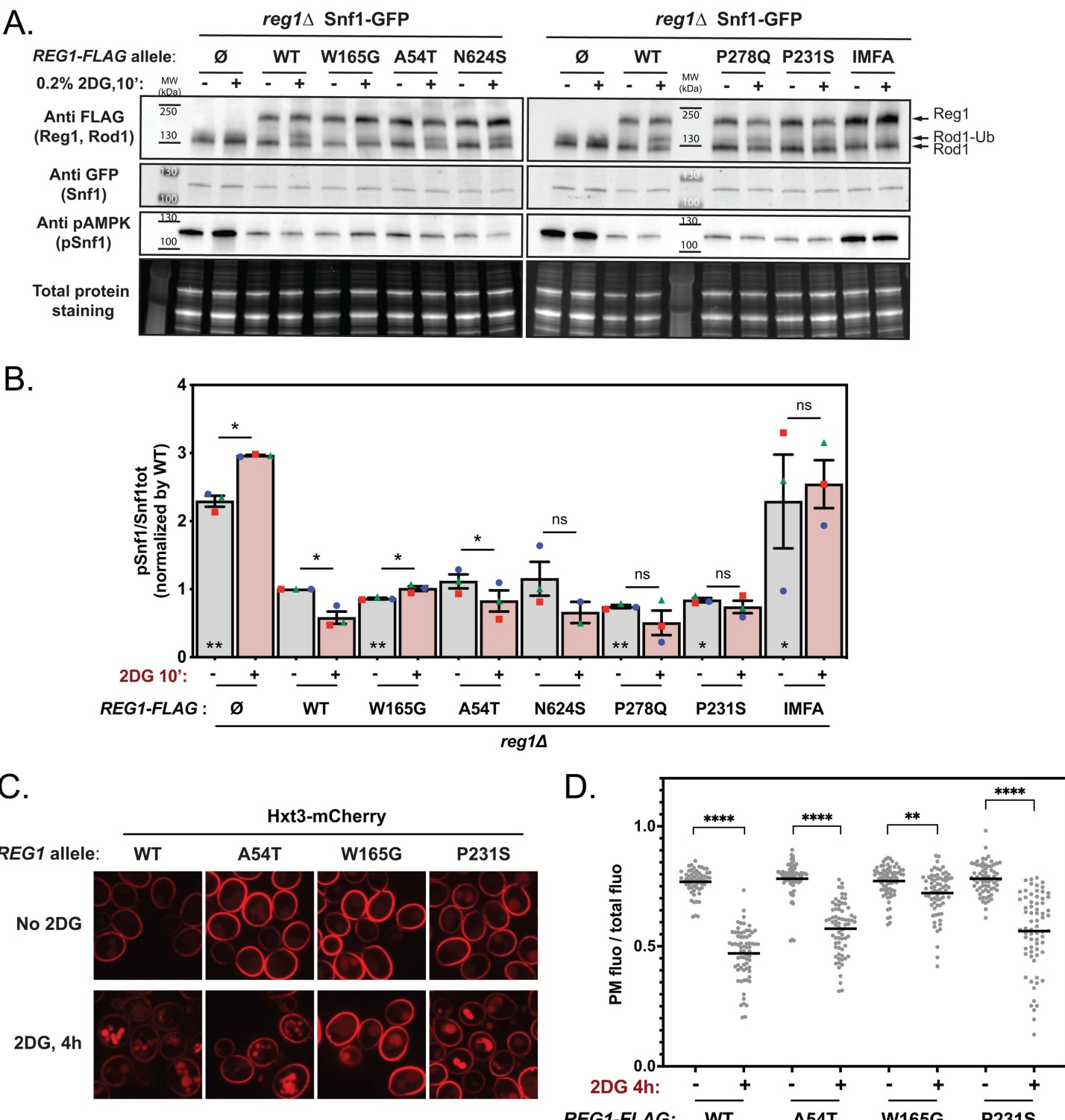

**Fig 8. Characterization of the 2DG response in Reg1 point mutants, focusing on Rod1 dephosphorylation and function. A.** Total protein extracts of cells from the mentioned genotype, expressing Rod1-FLAG and transformed or not with plasmid expressing the indicated Reg1-FLAG alleles, were prepared before and after 2DG addition for 10', and immunoblotted using anti-FLAG (Reg1 and Rod1), anti-GFP(Snf1) and anti-pAMPK (pSnf1) antibodies. **B.** Histogram representing Snf1 phosphorylation state (pSnf1/Snf1tot) in the mentioned strains, before or after 2DG treatment. * on top of

the bars = 0' vs. 10' 2DG treatment, * inside the bars, comparison of the 0' 2DG time point to the WT. $n = 3$, error bars = SEM, * indicate the *p-value* of statistical comparisons: *P < 0.05; **P < 0.01; ***P < 0.001; ****P < 0.0001. **C.** Cells expressing Hxt3-mCherry and WT or mutant Reg1-GFP were grown in a glucose-containing medium and observed by fluorescence microscopy before and after 2DG treatment for 150'. Scale bar, 5 µm. **D**. Quantification of the signal detected at the plasma membrane over the total cellular signal (*n* = 60 cells*)*.

hexose transporters is detrimental to cells in these conditions, possibly in relationship with its ability to detoxify 2DG. In line with this, deletion of *ROD1* caused 2DG resistance [50,52]. Therefore, Snf1 hyperactivity may trigger complementary 2DG resistance strategies. Overall, we found that the *reg1Δ* mutant, in which Snf1 is constitutively active, accumulates very low levels of 2DG6P after a short pulse of 2DG. This suggests that Snf1 hyperactivation leads to either a lack of 2DG transport inside the cells, its phosphorylation, or complete 2DG6P detoxification, or a combination of several of these effects. In line with these findings, *REG1* deletion was reported to decrease low-affinity hexose transporter expression and localization at the plasma membrane [56] as well as a higher expression of Dog2, a 2DG6P phosphatase [53].

An additional class of mutants that our study revealed are mutated in *REG1* but carry missense mutations that do not recapitulate the phenotypes obtained for a *reg1Δ* strain. First, whereas the , e.g.,*1Δ* strain displays a slow-growth in glucose medium, this was not the case for the missense mutants. Second, the latter were not hypersensitive to tunicamycin or selenite, which are drugs that are toxic to *reg1Δ* cells. Third, the glucose-mediated repression of *SUC2* or *DOG2* was not affected in these mutants. Finally, the phosphorylation of Snf1 or its targets Mig1 or Rod1 was not increased at steady state in these mutants compared to a WT strain. Overall, these data clearly demonstrate that despite being mutated in *REG1* and being resistant to 2DG, the mutants do not seem to display increased basal Snf1 signaling. Thus, SNF1 does not need to be constitutively active for yeast to become resistant to 2DG. A similar observation was made for mutant Reg1-P231L, analogous to the P231S mutation we report here, which caused 2DG resistance but seemed to have little consequences on Reg1 function [46].

The mutations we identified target conserved residues within conserved domains, however there is limited structural information available on Reg1. Truncation analyses using yeast two-hybrid revealed regions of Reg1 that are necessary for binding to its known partners, namely Glc7, Snf1 and 14-3-3 proteins [27,36,87,88]. However, binding to these interactants was not affected by the mutations we identified. Structural predictions [94] do not inform on a potential structure, since Reg1 is predicted to be mostly disordered [95,96], which is a common trait for PP1 regulatory proteins [83]. Bioinformatic analyses revealed the presence of a putative domain of unknown function (DUF1752) located in the Nt region of Reg1 (residues 173–204), in which fall several of the mutations we identified. This domain is present in the Reg1 paralogue Reg2, suggesting that it may be important for a function in Snf1 regulation, but more work will be required to assign it to a specific function. Thus, it is difficult to predict the structural consequences of the mutations we identified.

Of the many Reg1 mutants we obtained, three mutants were tested for 2DG6P content after a pulse of 2DG and the results indicate that they were able to phosphorylate 2DG to comparable levels as WT cells, highlighting again a difference with the *reg1Δ* mutant strain. This also indicates that these Reg1 mutants are not resistant because of lack of transport, phosphorylation, or increased detoxification, but rather that 2DG6P is tolerated and does not trigger a response that is detrimental to cell growth in these conditions. This is reminiscent of previous hypotheses that 2DG sensitivity could actually be caused by an excessive SNF1 activation rather than by the 2DG-induced metabolic blockade itself [46,52], and that limiting the magnitude of the response, either transcriptionally [46], at the level of endocytosis [50,56] or signaling (this study) is key for survival in the presence of 2DG. The inability of some mutants (e.g., Reg1-W165G) to inhibit SNF1 activity in response to 2DG might represent and advantage for growth in the presence of this toxic analogue. Alternatively, it is possible that in these mutants, 2DG6P is detoxified over time, allowing growth in these conditions.

Our study of five Reg1 mutants carrying mutations in various domains revealed heterogeneity in terms of their reaction to 2DG.

Intriguingly, two Reg1 mutants were not affected with regards to their ability to dephosphorylate Snf1, Mig1 or Rod1. At least one of these mutants, Reg1-A54T, also showed the typical vacuolar fragmentation observed in WT cells after 2DG

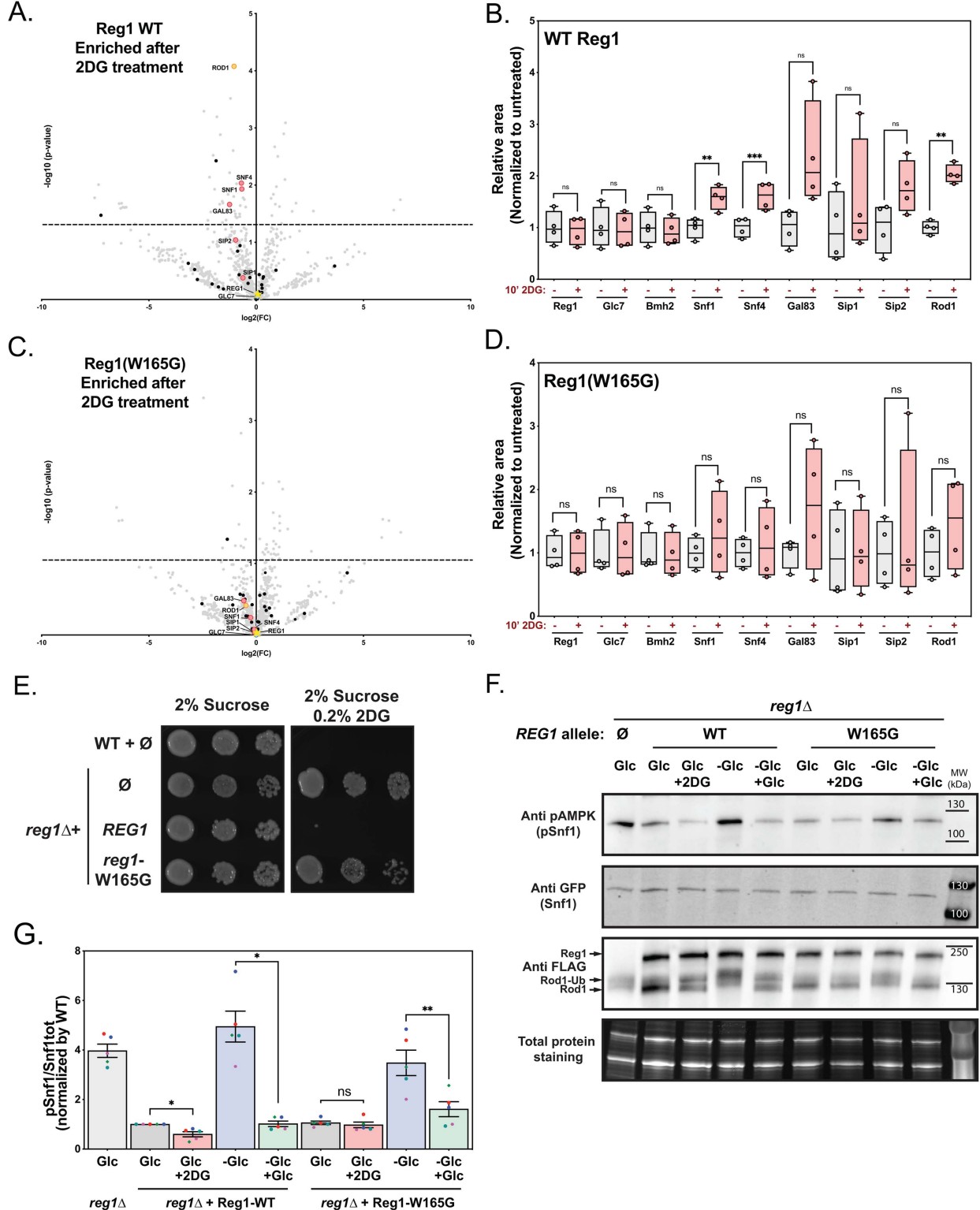

**Fig 9. The Reg1-W165G mutant is specifically affected for 2DG sensing. A,C**. Cells expressing WT or mutant Reg1-FLAG were grown to exponential phase overnight and treated with 2DG for 15'. Cells were lysed before and after treatment, Reg1-FLAG was immunoprecipitated and the immunoprecipitates were analyzed by quantitative, label-free proteomics. Volcano plots indicate proteins identified by mass spectrometry of WT or mutant

Reg1-FLAG, before or after 2DG treatment, with -log($p$ value) as a function of log2(untreated/treated ration). **B,D**. Box-plots comparing the relative abundance of 9 proteins measured by mass spectrometry before and after 2DG treatment (relative areas of the peaks, normalized to untreated sample for each protein). **E**. Growth assay of the indicated strains complemented or not with a centromeric plasmid allowing the expression of Reg1-Flag (WT or mutant) under the control of its endogenous promoter. Serial dilutions of yeast cultures were spotted on synthetic medium containing 2% sucrose, supplemented or not with 0.2% 2DG, and grown for 3 days at 30°C. **F**. Total protein extracts were prepared from cells of the indicated phenotype expressing Rod1-FLAG and WT or mutant Reg1. Cells grown in 2% glucose-containing medium were either treated with 2DG or resuspended in 0.05% glucose-containing medium (-Glc) for 10'. 2% glucose was then added to starved cells for 10'. The extracts were immunoblotted using anti-FLAG (Reg1 and Rod1), anti-GFP (Snf1) and anti-pAMPK (pSnf1) antibodies. **A-D**. n = 4, *p-value* of statistical comparisons: *$p < 0.05$; ** $p < 0.01$; ** $p < 0.001$; **** $p < 0.0001$. **G**. Quantification of Snf1 phosphorylation from panel **F**. $n = 3$, error bars: SEM, * indicate the *p-value* of statistical comparisons: **$p < 0.05$; ** $p < 0.01$.

treatment [50], and thus was indistinguishable from WT Reg1 on all the aspects that we checked, except for its ability to confer growth on 2DG. More work would be required to understand the basis of 2DG resistance in the context of these mutations, but this suggests that additional mechanisms are at stake.

Two mutants, instead, showed a defective dephosphorylation of Snf1 but only a partial defect for Rod1 and Mig1. Whereas data support a direct role of PP1 in Mig1 dephosphorylation [31,40], it remains unclear if this the case for Rod1 [60]. Indeed, Rod1 and Reg1 interact in yeast two-hybrid, but it could be indirect, and the high phosphorylation status of Rod1 observed in *reg1*Δ cells could be due to the high activity of Snf1 observed in this mutant [60]. However, we noticed that Mig1 and Rod1 phosphorylation were similarly affected in the Reg1 mutants studied. The fact that Rod1 and Mig1 can sometimes be dephosphorylated in the absence of Snf1 dephosphorylation (e.g., mutants Reg1-P231S and Reg1-P278Q) is in favor of Rod1 and Mig1 being direct targets of PP1. This also suggests that the Reg1-dependent dephosphorylation of Snf1 and Rod1/Mig1 may use different regions of the protein, or at least can be functionally separated. This is reminiscent of previous work showing that PP1 dephosphorylates Snf1 and Mig1 differently with respect to the glucose signal [31].

Finally, one mutant, Reg1-W165G, was strongly affected and 2DG treatment did not lead to the dephosphorylation of Snf1, Rod1 nor Mig1. This suggested that this mutation renders Reg1 activity towards these substrates insensitive to 2DG. Interestingly, Reg1-W165G was still able to dephosphorylate Snf1 in response to glucose, as would be expected from the fact that Snf1 activity is not constitutively high in presence of glucose. This is supported by quantitative proteomics indicating that the recruitment of WT Reg1 to Snf1 and Rod1 observed in response to 2DG no longer occurs for this mutant. Thus, the W165G mutation may affect 2DG sensing, perhaps by preventing binding to a metabolite or a conformational change in response to 2DG which would trigger a 2DG-induced recruitment of Reg1 on its substrates.

The nature of the signal that is perceived in presence of glucose and leads to SNF1 inhibition through PP1-mediated dephosphorylation is a long-standing question. The importance of glucose phosphorylation was already suggested, as the lack of glucose-phosphorylating enzymes prevents the glucose response [97]. The fact that 2DG, which is phosphorylated but not further metabolized, can substitute for glucose regarding Snf1 inhibition also supports a role for this metabolic reaction as a switch to trigger PP1 activity towards SNF1 [34]. Our results, showing that point mutations in Reg1 block signaling from 2DG metabolism but not from glucose metabolism, support the hypothesis that Reg1 functions as a metabolic sensor. Specifically, Reg1 may directly interact with a metabolite to mediate its recruitment to the SNF1 complex and promote its dephosphorylation in response to nutrient cues.

## Materials and methods

### Yeast strain construction and growth conditions

All yeast strains used in this study derive from the *Saccharomyces cerevisiae* BY4741 or BY4742 background and are listed in S3 Table. Apart from the mutant strains obtained from the yeast deletion collection (Euroscarf), all yeast strains were constructed by transformation with the standard lithium acetate–polyethylene glycol protocol using homologous recombination and verified by polymerase chain reaction (PCR) on genomic DNA prepared with a lithium acetate

(200 mM)/SDS (0.1%) method [98]. Yeast cells were grown in YPD medium (2%) or in SC medium [containing yeast nitrogen base (1.7 g/liter; MP Biomedicals), ammonium sulfate (5 g/liter, Sigma-Aldrich), the appropriate drop-out amino acid preparations (MP Biomedicals), and 2% (w/v) glucose, unless otherwise indicated]. Precultures were incubated at 30°C for 8 hours and diluted in the evening to reach mid-log phase (OD$_{600}$ = 0.3-0.5) the next morning. 2DG (Sigma) was added to mid-log phase yeast cultures grown overnight to final concentrations of 0.2% (w/v) and cells incubated for the indicated times. For starvation, mid-log phase cells were harvested and resuspended in a medium containing 0.05% glucose for 10' minutes. Glucose (2% final concentration) or 2DG (0.2% final concentration) were added to starved cells.

## Plasmid construction

All plasmids used in this study are listed in S4 Table.

Plasmids expressing Reg1 wild-type or the mutant Reg1 tagged with FLAG were obtained starting from pSL559 (pRS313, p$_{ROD1}$-ROD1-FLAG-t$_{ROD1}$) [50]. The plasmid was digested with SalI and NcoI to remove p$_{ROD1}$-ROD1 which was substituted by WT or mutant Reg1 amplified by PCR from gDNA of WT (BY4741) or the spontaneous mutants isolated which carried the desired mutation. The assembly of the vector and the insert was done with the NEBuilder HiFi DNA Assembly Cloning Kit (NEB) as per the manufacturer's instructions.

Plasmids expressing Snf1, Snf4 and Gal83 (either WT or mutant) fused to mCherry were obtained by cloning a PCR product from gDNA (including 500 bp of promoter) in frame with mCherry at SacI/BamHI sites in pSL21 (pRS416, p$_{GPD}$-mcs-mCherry-t$_{CYC1}$, [99]) in place of the p$_{GPD}$ promoter.

## Isolation of spontaneous 2DG resistant mutants, complementation analysis and phenotypic characterization

WT (BY4742, Mat α) cells were streaked on a YPD plate and incubated overnight. Ten single colonies were then re-streaked in 10 different YPD plates which were used to establish 10 liquid cultures. This was done to increase the diversity of mutations identified. On average 9x10$^6$ were plated on each SC + 0.2% 2DG plate. These were grown at 30°C for 6 days. 195 of the 2DG resistant clones obtained were re-streaked on YPD rich medium to isolate single colonies and on SC + 0.2% 2DG to confirm the resistance.

**Phenotypic characterization.** Cells from isolated colonies were used to set up 200 µL YPD liquid culture in 96-well plates, which were incubated at 30°C for 1 day. Then, cells were pinned into SC-agar medium or SC-agar medium supplemented with 0.2% 2DG, 0.5 µg/mL Tunicamycin or 200 mM sodium selenite using a robot for high throughput screening (Rotor HDA, Singer Instruments). The plates were incubated at 30°C for three days before being scanned using a flatbed scanner (EPSON). Iodine staining for the detection of glycogen-accumulating colonies was performed inverting YPD plates over di-iodine crystals for 2 minutes [modification of 100]. Plates were scanned with a flatbed scanner.

**Complementation groups analysis.** Cells from isolated colonies were used to set up 200µL liquid culture in 96-well plates, which were incubated at 30°C for 1 day. Diploids were obtained by pinning them on SC-agar medium plates depleted of methionine and lysine, onto which mutants from the opposite mating type (a) were previously spread, using the ROTOR HDA robot. After 1 day of incubation at 30°C, 200 µL liquid cultures of the obtained diploids were established in 96-well plates and incubated for 1 day at 30°C. The cells were then pinned on SC-agar medium depleted of methionine and lysine either containing or not 0.2% 2DG, and grown at 30°C for 3 days before scanning them.

## Quantification of growth on plates and assignation of a mutant to a complementation group

Colonies on plate were quantified using ImageJ software (NIH) and the "plate analysis jru v1" plug-in from the Stowers Institute. The following settings have been used: #of spots:384; XY ratio:1,5; spot radius:25; #x replicates:2; #y replicates:2; Circ background function activated; Circ background Stat: Avg. Data was analyzed on Excel. Heatmaps were created by applying conditional formatting to the data, after normalization to the WT in each condition. The assignment of a mutant to a specific complementation group was based on the comparison between the growth of a mutant and the

corresponding control strain. If the growth was higher than the control strain minus 20% (to allow for some variability), then a mutant was considered part of a complementation group. A mutant was considered dominant when it grew at least 2-fold more with respect to the WT.

### Whole genome re-sequencing

1.5µg of gDNA extracted via the Phenol:Chloroform:Isoamyl Alcohol-based protocol [46], and whole genome re-sequencing was performed at the BGI (Beijing Genomics Institute, Hong-Kong). A PCR-based method was used for the construction of the library. DNA Nanoball Sequencing (DNBSEQ) was performed. The sequencing coverage was higher than 99% in all cases and the sequencing depth was around 98X. The sequence was compared with a reference sequence (Genome accession: txid4923) [101,102] and the mutations were detected by mapping the WT reads to the reference. The differential variants were filtered by quality (vcf QUAL>1000) and manually inspected through IGV for validation [103].

### Total protein lysates with TCA

Yeast cells were grown to the mid-log phase ($OD_{600}$ of 0.4-0.6) in SC medium, depleted of one amino acid to select for auxotrophy when required. For each protein sample, 1.4 ml of culture was incubated with 100 µl of 100% TCA for 10 min on ice to precipitate proteins, centrifuged at 16,000 xg at 4°C for 10 min, and vortexed for 10 min with glass beads. Lysates were transferred to another 1.5-ml tube to remove glass beads and centrifuged for 5 min at 16,000 g at 4°C, supernatants were discarded, and protein pellets were resuspended in Laemmli sample buffer [50 mM tris-HCl (pH6.8), 100 mM dithio-threitol, 2% SDS, 0.1% bromophenol blue, and 10% glycerol, complemented with 50 mM tris-base (pH 8.8)] (50 µl/initial $OD_{600}$). Samples were then denatured at 95°C on a heat block for 10 minutes.

### Immunoprecipitation of Reg1-FLAG in native condition

150-200 mL cultures were grown to an $OD_{600}$ of 0.4-0.6 (mid-log phase). Cells were added to 10 uL/OD of IP lysis buffer [36] (50 mM HEPES (pH 7.5), 150 mM NaC1, 0.5% Triton X-100, 1 mM dithiothreitol, 10% glycerol and Protease inhibitor cocktail (Sigma-Aldrich)) and lysed with acid-washed glass beads by vortexing 4 times for 30 seconds at 4°C, with 1 minute on ice in between. The lysate was centrifuged for 5 min at 3000 xg at 4°C to remove cellular debris. The supernatant was added to agarose beads coated with anti-FLAG M2-Agarose from mouse (Sigma) and mixed for 1 hour on a wheel at 4°C. After centrifugation, beads were washed on a spin column three times with 1 mL cold wash buffer (lysis buffer without protease inhibitors). For immunoblotting, the final elution was done in 1X Laemmli Buffer (1 µL/ initial $OD_{600}$). Eluates were then denatured at 95°C on a heat block for 10 minutes. An aliquot of the supernatant was taken before adding it to the beads (inputs). An equal amount of 2X Laemmli Buffer was added before denaturation at 95°C on a heat block for 10 minutes. Volumes corresponding to 0.2 (input) and 10 (IP) $OD_{600}$ of cells, respectively, were loaded on gel. For mass spectrometry analysis, the beads were resuspended in $H_2O$ (1 µL/ initial $OD_{600}$) and treated for MS analysis.

### Immunoblotting

After denaturation, protein samples were loaded on SDS–polyacrylamide gel electrophoresis (PAGE) gels (4–20% Mini-PROTEAN TGX Stain-Free, Bio-Rad). After electrophoresis for 30–45 min at 200V, total proteins were visualized by in-gel fluorescence using a trihalo compound incorporated in SDS–PAGE gels (stain-free TGX gels, 4–20%; Bio-Rad) after 45 sec UV-induced photoactivation using a ChemiDoc MP imager (BioRad), serving as a loading control. Gels were transferred on nitrocellulose membranes for 60 min (100V) in a liquid transfer system (Bio-Rad). Membranes were blocked in Tris-buffered saline solution containing 0.5% Tween-20 (TBS-T) and 2% fat-free milk for 30 min and incubated for at least 2 hours with the corresponding primary antibodies. Membranes were washed 3x10 min in TBS-T and incubated for at least an hour with the corresponding secondary antibody (coupled with horseradish peroxidase). Membranes were then washed again 3 x10 min in TBS-T and incubated with SuperSignal West Femto reagent (Thermo). Luminescence signals

were acquired using a ChemiDoc MP (BioRad). Primary and secondary antibodies used in this study as well as their dilutions are listed in S5 Table.

## Mass spectrometry

**Sample preparation.** Beads from pulldown experiments were incubated overnight at 37°C with 20 µL of 50 mM $NH_4HCO_3$ buffer containing 1 µg of sequencing-grade trypsin. The digested peptides were loaded and desalted on evotips (Evosep) according to the manufacturer's procedure.

**LC-MS/MS acquisition.** Samples were analyzed on a timsTOF Pro 2 mass spectrometer (Bruker Daltonics) coupled to an Evosep one system (Evosep) operating with the 30SPD method developed by the manufacturer. Briefly, the method is based on a 44-min gradient and a total cycle time of 48 min with a C18 analytical column (0.15 x 150 mm, 1.9µm beads, ref EV-1106) equilibrated at 40°C and operated at a flow rate of 500 nL/min. H2O/0.1% FA was used as solvent A and ACN/ 0.1% FA as solvent B. The timsTOF Pro 2 was operated in PASEF mode (Meier et al., 2015) over a 1.3 sec cycle time. Mass spectra for MS and MS/MS scans were recorded between 100 and 1700 m/z. Ion mobility was set to 0.75-1.25 V·s/cm2 over a ramp time of 180 ms. Data-dependent acquisition was performed using 6 PASEF MS/MS scans per cycle with a near 100% duty cycle. Low m/z and singly charged ions were excluded from PASEF precursor selection by applying a filter in the m/z and ion mobility space. The dynamic exclusion was activated and set to 0.8 min, a target value of 16000 was specified with an intensity threshold of 1000. Collisional energy was ramped stepwise as a function of ion mobility.

**Data analysis.** MS raw files were processed using PEAKS Online X (build 1.8, Bioinformatics Solutions Inc.). Data were searched against the Saccharomyces cerevisiae SwissProt database (release 2022_01, total entry 6062 entries). Parent mass tolerance was set to 20 ppm, with fragment mass tolerance at 0.05 Da. Specific tryptic cleavage was selected and a maximum of 2 missed cleavages was authorized. For identification, the following post-translational modifications were included: oxidation (M), acetylation (Protein N-term), phosphorylation (STY) and deamidation (NQ) as variables and half of a disulfide bridge (C) as fixed. Identifications were filtered based on a 1% FDR (False Discovery Rate) threshold at both peptide and protein group levels. Label free quantification in Fig 9 was performed using the PEAKS Online X quantification module, allowing a mass tolerance of 20 ppm, a CCS error tolerance of 0.05 and Auto Detect retention time shift tolerance for match between runs. Protein abundance was inferred using the top N peptide method and TIC was used for normalization. Multivariate statistics on proteins were performed using Qlucore Omics Explorer 3.8 (Qlucore AB). A positive threshold value of 1 was specified to enable a log2 transformation of abundance data for normalization, i.e., all abundance data values below the threshold will be replaced by 1 before transformation. The transformed data were finally used for statistical analysis, i.e., evaluation of differentially present proteins between two groups using a Student's bilateral t-test and assuming equal variance between groups. A *p-value* better than 0.01 was used to filter differential candidates. Protein abundance plots were calculated as follows: $(\log_2(X)+\log_2(Y))/2$, where X is the average of the "Area" measured for a protein found in the sample from cells not exposed to 2DG and Y is the average of the "Area" measured for a protein found in the sample after 2DG treatment.

## Quantification of Western-blots

Western blot quantifications were performed with ImageJ (NIH). The integrated density of the band was measured. Figs 2C, 5C-D, and 6B: the integrated density of the band representing the protein of interest was divided by the integrated density of total proteins. Results were normalized to the WT. Fig 6C: the integrated density of the band representing Snf1 or Glc7 or 14–3-3s was divided by the integrated density of the Reg1 band. Results were normalized to the WT. Figs 7A, 7C, 8A, 8B, 9F and 9G: the integrated density of the band representing pSnf1 was divided by the integrated density of the band for total Snf1. Results were normalized to the WT at t = 0'.

## Quantification of Mig1 shift

The quantification of the shift of the Mig-FLAG bands was done using the ImageQuant TL (IQTL) analysis software, on the images of membranes incubated with anti-FLAG antibodies. The lanes and bands were automatically detected and then minimal manual adjustments performed. The distance between the top of the gel and the region of the band where maximal intensity was detected was measured. Data was analysed in Excel and the relative distance between two bands (same strain, t = 0' vs. t = 10' 2DG treatment) was calculated and plotted.

## Beta-galactosidase assays

Cultures were established from overnight saturated cultures at an initial concentration of 0.1 OD/mL and incubated at 30°C for 6 hours. The OD was measured and 25 and 50 $\times 10^{-3}$ $OD_{600}$ equivalent of cells per strain were taken to perform the measurements. The assay was carried out using Gal-Screen β- Galactosidase Reporter Gene Assay System for Yeast or Mammalian Cells (Thermofisher) following the instruction provided by the manufacturer. Chemiluminescence was measured for 1 hour with a 1-min interval using a Spark multimodal plate reader (TECAN). The same strain was measured four times in each experiment.

## 2DG-6-phosphate assays

Cells were grown overnight in SC medium until mid-log phase. 2DG was added into the culture to a final concentration of 0.2% for 15'. 1 OD equivalent of cells were taken, immediately placed on ice, centrifuged at 4°C for 4' at 16,000g, washed once with cold PBS and lysed using Y-PER Yeast Protein Extraction Reagent (ThermoFisher Scientific) as previously described [50]. Briefly, 1 OD equivalent of cells was resuspended in 100μL of Y-PER buffer, incubated for 20' at 30 °C, and centrifuged for 4' at 16100g at room temperature. The supernatant (between 0.0125 and 0.0625 $OD_{600}$ equivalents) was used for the assay and diluted in Y-PER when needed. The assay was performed using the 2DG uptake measurement kit (Cosmo Bio USA, Carlsbad, CA, USA;, Ref. CSR-OKP-PMG-K01H) following the manufacturer's protocol adapted for a total reaction volume of 100 μL. Absorbance at 420 nm was measured every minute for 1 hour on a SpectraMax M2 Microplate Reader (Molecular Devices). Measures taken between 20 and 35 minutes (linear range) were taken into account to compare the slopes. Each sample was assayed at least twice in each experiment and at least three independent experiments were performed.

## ATP FRET sensor experiments

ATP FRET assays were performed as described previously [50]. Yeast transformed with the plasmid pDRF1-GW yAT1.03 (pSL608) (a gift from Bas Teusink: Addgene #132781) [89] and, when mentioned, with a Reg1-FLAG plasmid were grown overnight in SC-HIS-URA medium to reach an $OD_{600}$ of 0.3-0.7 in the morning. Cells were centrifuged at 3000 x*g* for 5 minutes and the cell pellets were resuspended in 20mM MES pH = 6 to a final OD of 0.6. 180μl of the cell suspension were distributed into a 96-well black PS microplate with flat bottom (Greiner). Fluorescence was measured with a plate reader (Spark, TECAN) set at 30°C before the experiment was started. Cells were excited at 438 nm and the emission intensities at 483 nm (ymTq2Δ11) and 593 nm (tdTomato) were collected. A first measurement for 2 cycles was performed to establish a baseline (each cycle is around 90 seconds with 10 seconds shaking before each measurement of the fluorescence). The cycle was then paused to inject 20μl of the 2DG (final concentration of 0.2%) solutions using a multichannel pipette, and the plate rapidly reloaded in the reader to restart measurements for another 18 cycles. Data was collected and analyzed in Excel. Emission intensity values of yeast transformed with an empty plasmid were subtracted from emissions values at 438 nm and 593 nm for each measurement at the different glucose concentrations. The emission intensity ratio for dTomato (593 nm) over ymTq2Δ11 (483 nm) was calculated. Emission ratios were normalized to the average of the three initial ratio values before glucose addition. All analyses were repeated (with three technical replicates) at least three times independently.

## Drop tests

Yeast cells grown in liquid-rich or SC medium for at least 6 hours at 30°C were adjusted to an optical density (600 nm) of 0.5. Serial 10-fold dilutions were prepared in 96-well plates and spotted on plates containing rich or SC medium containing 2% (w/v) agar and, when indicated, 2DG [0.05 or 0.2% (w/v)], Tunicamycin (0,5μg/mL) or Selenite (200mM). Plates were incubated at 30°C for 3–4 days before taking pictures using the Colorimetric function of the Chemidoc MP imager (BioRad).

## Microscopy

Images presented are spinning disk confocal mid-sections. Imaging was carried out at room temperature (22–25°C) on an inverted spinning-disk confocal microscope equipped with a motorized stage and an automatic focus (Ti-Eclipse, Nikon, Japan), a Yokogawa CSUX1FW spinning unit, and an EM-CCD camera (ImagEM-1K, Hamamatsu Photonics, Japan), or a Prime BSI camera (Photometrics). Images were acquired with a 100X oil-immersion objective (CFI Plan Apo DM 100X/1.4 NA, Nikon).

## Quantification of microscopy images

Images were treated in ImageJ. Ellipses were drawn around each cell (E1) and its cytosol (E2), and in the background nearby (E0). The integrated density (IntDens) and area (Area) were measured for E1 and E2, and the values obtained were corrected for background by subtracting the pixel mean value (Mean) of E0: $IntDensCorr_{E1} = IntDens_{E1} - (Area_{E1} \times Mean_{E0})$ and $IntDensCorr_{E2} = IntDens_{E2} - (Area_{E2} \times Mean_{E0})$. The PM/total ratio was calculated as $(IntDensCorr_{E1} - IntDensCorr_{E2})/IntDensCorr_{E1}$. Quantification was achieved for 40 cells in each experiment, and 3 independent experiments were quantified.

## Statistical analysis

Mean values were calculated from at least three independent measurements and were plotted with error bars representing SEM. Unless differently stated, statistical significance was determined using a t-test for paired variables assuming a normal distribution of the values using GraphPad Prism v.9.

## Supporting information

**S1 Data.  Raw data used in the figures.**
(XLSX)

**S1 Fig.  Flow chart representing the steps of mutant characterization.** The number of mutants at each step is indicated (*n*). Numbers in blue/ italics represent the ID of the mutants (see S1 Table).
(EPS)

**S2 Fig.  Reg1-FLAG is functional.** Serial dilutions of cultures of the WT, *reg1Δ*, or *reg1Δ* transformed with a Reg1-FLAG plasmid were spotted on a synthetic medium (histidine drop-out) containing 2% glucose and supplemented with 0.2% 2DG, 2 mM selenite or 0.5 μg/mL tunicamycin and grown for 3 days at 30°C.
(EPS)

**S1 Table.  Results of the complementation group analysis of the 2DG-resistant mutants obtained in the screen.**
(DOCX)

**S2 Table.  Quantification of growth on various drugs of the 2DG-resistant mutants obtained in the screen.**
(DOCX)

**S3 Table. Strains used in this study.**
(DOCX)

**S4 Table. Plasmids used in this study.**
(DOCX)

**S5 Table. Antibodies used in this study.**
(DOCX)

## Acknowledgments

The authors wish to thank Marion Lima, Zoe Zilber and Karolína Černá for assistance in endocytosis experiments and sequencing of mutants, Dr Matthieu Sanial for help with immunoblot quantification, Dr Alexandre Soulard (Université Claude Bernard – Lyon 1, FR) for helpful discussions and criticism and Jay Unruh (Stowers Institute for Medical Research, Kansas City, MO, USA) for the plate analysis ImageJ Plug-in. We thank Nicolas Minc (Institut Jacques Monod, Paris, FR) for the use of his spinning-disk microscope.

## Author contributions

**Conceptualization:** Alberto Ballin, Véronique Albanèse, Guillaume Chevreux, Sébastien Léon.

**Data curation:** Alberto Ballin, Anne Friedrich, Sébastien Léon.

**Formal analysis:** Alberto Ballin, Véronique Albanèse, Véronique Legros, Guillaume Chevreux, Agathe Verraes, Anne Friedrich, Sébastien Léon.

**Funding acquisition:** Alberto Ballin, Guillaume Chevreux, Sébastien Léon.

**Investigation:** Alberto Ballin, Véronique Albanèse, Samia Miled, Véronique Legros, Agathe Verraes, Sébastien Léon.

**Methodology:** Alberto Ballin, Véronique Albanèse, Samia Miled, Véronique Legros, Agathe Verraes, Anne Friedrich.

**Project administration:** Sébastien Léon.

**Resources:** Alberto Ballin, Véronique Albanèse, Agathe Verraes.

**Supervision:** Guillaume Chevreux, Sébastien Léon.

**Validation:** Alberto Ballin, Véronique Albanèse, Samia Miled, Guillaume Chevreux, Agathe Verraes, Anne Friedrich, Sébastien Léon.

**Visualization:** Alberto Ballin, Véronique Albanèse, Samia Miled, Véronique Legros, Guillaume Chevreux, Sébastien Léon.

**Writing – original draft:** Alberto Ballin, Véronique Albanèse, Guillaume Chevreux, Sébastien Léon.

**Writing – review & editing:** Sébastien Léon.

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
