## [Decision Letter · Decision Letter 0]

29 Jul 2025

PGENETICS-D-25-00740

A genetic screen reveals a key role for Reg1 in 2-deoxyglucose sensing and yeast AMPK inhibition

PLOS Genetics

Dear Dr. Leon,

Thank you for submitting your manuscript to PLOS Genetics. Your manuscript “A genetic screen reveals a key role for Reg1 in 2-deoxyglucose sensing and yeast AMPK inhibition” has been evaluated by three experts. You will see in the enclosed reviews that the reviewers agree that your studies are thorough and of high quality although each of the reviewers raised numerous questions and suggested multiple, primarily editorial, modifications. However, in both the written reviews and in comments to the editors each of the reviewers raised concerns as to the novelty of your conclusions. PLOS Genetics publishes "original research that clearly demonstrates novelty, importance to a particular field, and biological significance". So, the reviewers’ comments raise the concern that your manuscript may be more appropriate for a different journal. On the other hand, we suspect that you and your co-authors may have not convincingly communicated the novel findings of your studies and how they advance the function of Reg1 and/or glucose repression. Thus, we invite you to submit a revised version of the manuscript that addresses the points raised during the review process. If you elect to submit a modified version of Ballin et al. you must satisfactorily address the "novelty" issue in your revised manuscript.

Please submit your revised manuscript within 30 days Aug 28 2025 11:59PM. If you will need more time than this to complete your revisions, please reply to this message or contact the journal office at plosgenetics@plos.org. Please include the following items when submitting your revised manuscript:

We look forward to receiving your revised manuscript.

Kind regards,

Anita K. Hopper

Academic Editor

PLOS Genetics

Geraldine Butler

Section Editor

PLOS Genetics

Aimée Dudley

Editor-in-Chief

PLOS Genetics

Anne Goriely

Editor-in-Chief

PLOS Genetics

**Comments to the Authors:**

Reviewer #1: Cells can adapt to environmental changes by evolutionarily conserved regulatory pathways. Critically important pathways control the response to fluctuating nutrient levels. One of the best studied pathways of nutrient control is the “glucose repression” pathway that allows yeast cells to switch to growth on poor carbon sources when the major source of nutrients, glucose, becomes limiting. A highly studied kinase, Snf1, homologous to AMPK in humans, controls the de-repression of glucose repressed genes. Surrounding Snf1 are a suite of regulatory proteins whose functions are well known. These include a transcriptional repressor, Mig1, regulatory subunits (Snf4), activating kinases (Elm1 and others), and an essential phosphatase complex (Glc7 and Reg1). Here, a new classical genetic screen (combined with a powerful pinning robot) was performed to identify new regulatory features of this signaling circuit. Mutants resistant to 2DG, a non-metabolizable glucose homolog, were uncovered. Complementation analysis followed by sequencing identified new dominant alleles of key regulatory genes and a large number of recessive alleles. These alleles were confirmed by sequencing and plasmid complementation, and overlaid onto the structures of the protein complexes. A large body of literature was also referred to in order to put the mutations into context. As a result of this effort, a new collection of missense mutations in the Glc7 regulatory subunit Reg1 were identified and characterized. These mutants were resistant to 2DG but did not show relief of other glucose repression phenotypes characteristic of cells lacking Reg1. The mutants were not defective for interaction with Glc7 or 14-3-3 proteins, but were defective in relaying the 2DG signal. The mutants in this class had different phenotypes in various aspects of Hxt transporter internalization and dephosphorylation of the arrestin, Rod1. One mutant W165G when analyzed by quantitative proteomics showed altered protein interactions with Rod1, reduced de-phosphorylation of Snf1, and differences in vacuolar fragmentation.

This is a well thought out, well written, and carefully executed study. A lot of work was done, and much of the model of the Glc7-Reg1-Snf1-Mig1 pathway was verified. The paper was relatively easy to read and understand. It is clear that a new aspect of Reg1 protein was identified that is involved in glucose control. There are a few important questions to address and some minor suggestions.

Main points

The major question that comes from reading the paper is what is this new function for Reg1? The authors claim that they have identified a “novel class of missense mutations”; however, their mutations fell into multiple phenotypic classes.

In the abstract, is the conclusion is made that “that 2DG toxicity primarily results from aberrant Snf1 inactivation rather than a direct metabolic disturbance.” What evidence support this conclusion?

Minor points

The introduction was well written. However, a simple model at the beginning might help the reader orient themselves as to the proteins involved.

Figures 5 and 6, there is no mention of statistical tests to show differences. Presumably these tests will be straightforward and will help the reader know which findings are significant and which are not (e.g., Figure 2C).

There are two 14-3-3 proteins, Bmh1 and Bmh2, which one was tested? Is it possible that there is a difference in the interaction with the other Bmh protein?

“The expression level was not drastically affected by the mutations (Fig 6B), except for the IMFA mutant which consistently showed a twofold, in agreement with previous observations [85].” Add “[twofold] reduction in protein levels”

Some effort on the figures could make them easier to interpret for the reader. For example:

a. Figure 1C, what does CG refer to?

b. Figure 2A, the order is Snf1, Gal83, Snf4, but in Figure 2B, the order is Snf1, Snf4, Gal83. Can the order of panel 2A be moved around to match?

c. Can panel 2E be given a title for the protein being modeled?

d. Perhaps the letters referring to the panels could be made bold and placed outside of the data in the panels?

e. In my version, something is wrong with the error bar in Fig. 7D.

Reviewer #2: This manuscript describes a large-scale screen for spontaneous mutations that confer resistance to the glucose analog, 2-deoxyglucose. The major finding is summarized in the manuscript title where they state that the Reg1 protein plays a key role in 2DG resistance and AMPK inhibition. The strengths of this paper are the thoroughness and scientific rigor of the study. The major weakness of this paper is the lack of novelty. The idea that Reg1 plays a key role in glucose repression and AMPK inhibition has been reported multiple times by many authors going back more than 30 years (Dombek et al 1993; Tu can Carslon 1995). The identification of Reg1 and Hxk2 in screens for 2DG resistance were first reported by Neigeborn and Carlson in 1987 and have been repeated multiple times in a number of labs, including the Leon lab. The big question is what is new here?

Major points

1. The major claim of novelty in this paper is that they have uncovered a new class of reg1 missense mutants. The prototype, Reg1-W165G fails to impact Snf1 basal activity. For comparison, reg1∆ cells have constitutively active Snf1 kinase while the reg1-W165G allele does not. The known functions of the Reg1 protein are binding to Glc7, Snf1 and Bmh1/2, Glc7 substrate recognition (Snf1, Mig1, Rod1, etc) and maintenance of glucose repression. The authors do not adequately show the W165G mutant differs from previously identified Reg1 missense alleles in any of these functions. For instance, the basal level of Snf1 phosphorylation is elevated in reg1∆ cells but not in W165G. However, it is not elevated in in the other missense alleles.

2. The authors have an assay that directly measures 2DG-6P in cell extracts (Fig 7D and Laussel 2022). One important finding is that reg1 delete mutants do not accumulate 2DG-6P while the missense alleles do. The idea that resistance to 2DG is independent of 2DG-6P accumulation challenges the prevailing model in which 2DG sensitivity is driven by 2DG‑6P accumulation. Why do reg1 delete strains fail to accumulate 2DG-6P? Why do missense alleles accumulate 2DG-6P but still confer resistance? Further the authors do not present the 2DG‑6P accumulation data in the newly characterized hxk2 alleles. Do any of their hxk2 alleles distinguish between glucose and 2DG?

3. The handling of screen “hits” feels inconsistent and difficult to track. A concise flow chart (or summary table) outlining each decision point would greatly improve clarity. Screen progression: show how 195 initial mutants were reduced to 157 by complementation with WT HXK2, REG1, or GLC7. Unassigned mutants (n = 38): list the genes in which these mutations occur and explain why their causative role in 2DG resistance was not confirmed (e.g., whole‑genome sequencing not pursued, technical limitations, etc.). Focused subsets: specify the rationale for choosing the 17 REG1 alleles for detailed analysis.

Minor points.

a. Some figures are not accessible to people with color vision issues. For instance, a red-green heat map scale (Fig 5A) and graphs with colors only (no symbols in the lines; Fig 7E).

b. Fig 4B shows Class I and Class II alleles. Fig 4C shows mutations in Reg1 with missense alleles on top and nonsense alleles on bottom. What are M1* and M1R? This is not explained in the legend.

c. Discussion mentions that the W165G mutant was “still able to dephosphorylate Snf1 in response to glucose” (p.19, line 461). Where is that shown?

d. Fig 7D has a misplaced error bar probably belonging to P231S

e. Fig 9 legend: “Wells were lysed” should be Cells.

Reviewer #3: This is an interesting paper which provides evidence for a crucial role of Reg1 in the initial glucose repression mechanism, based on 2DG toxicity/signaling experiments. The specific REG1 mutant alleles will be very useful for the further elucidation of the initial mechanisms of glucose sensing in the glucose repression pathway.

Other comments

Abstract

Please use the term repression only for downregulation at the transcriptional level, repression of a gene. In this sentence:

Among the latter, the Protein Phosphatase 1 (PP1) complex Reg1/Glc7 plays a critical role in repressing Snf1 activity under glucose-rich conditions.’ replace repressing by inhibiting.

The abstract is sometimes confusing because the authors shift between 2DG-induced glucose repression and 2DG-induced toxicity, without addressing the (possible) relationship between the two. There is nothing mentioned in the abstract how 2DG toxicity is related to 2DG-induced glucose repression. Is the latter part of the toxicity mechanism ? If yes, was this already known ?

For instance:

We describe a novel class of REG1 missense mutations, including reg1-W165G, that maintain normal basal Snf1 activity but fail to mediate 2DG-induced Snf1 inhibition, rendering cells effectively immune to 2DG toxicity.

Line 99 ‘On the other hand, deletion of HXK2 does not lead to major transcriptional changes [48].’

This is a weird statement since deletion of HXK2 eliminates glucose repression of many genes.

Line 104 ‘This response was not observed upon deletion of HXK2, which we proposed to be the main 2DG-phosphorylating enzyme in vivo.’ Would you not expect the opposite effect since glucose phosphorylation by hexokinase should cause the same effect of 2DG phosphorylation ?

Line 105 ‘Thus, 2DG phosphorylation, i.e. 2DG-6-phosphate itself or a derived metabolite, is central to trigger Snf1dephosphorylation.’ There is also the possibility that active hexokinase itself in some way triggers dephosphorylation of Snf1 as a regulatory protein.

Line 109 Up to this point the authors have only dealt with the glucose repression mechanism buy which glucose phosphorylation could cause inhibition of SNF1 phosphorylation. Now they will start a screen of 2DG resistant mutants. Is there any evidence that a 2DG resistant mutant would have altered glucose-induced inhibition of Snf1 ? 2DG tolerance can also be caused for instance by higher activity of the DOG1 and DOG2 phosphatases. This aspect should be dealt with.

Line 114 ‘The second one is caused by an increased basal

115 SNF1 activity, either through gain-of-function mutations in all three subunits of AMPK (SNF1, SNF4, GAL83) as previously reported [54] or loss-of-function mutation in the PP1 phosphatase genes, REG1 and GLC7 [45, 50-52].’ Hence, does higher respiration activity overcome 2DG toxicity ? But: Line 118: ‘probably through detoxification mechanisms’: what mechanisms ? Higher activity of DOG1 and DOG2 ?

Line129 ‘consolidate the idea that 2DG toxicity is not due to its direct metabolic effects but rather to an excessive response such as SNF1 inactivation which, when dampened, allows 2DG resistance.’ At first sight this is correct, but if higher Snf1 activity causes higher activity of the DOG1 and DOG2 2DG phosphatases, the idea is no longer correct. In that case it is basically the same as reducing Hxk2 activity, it would just reduce the accumulation of 2DG.

Line 133 ‘To gain insights into how 2DG is sensed and how it regulates AMPK activity, we performed a 134 genetic screen to isolate 2DG-resistant mutants in glucose-containing medium.’ This is the same issue: the authors link 2DG sensing for initiation of glucose repression to 2DG toxicity. What is the evidence that the two are related ? This should be evidenced more explicitly.

Line 169 ‘We also identified a mutation in TPS2 (clone #5.21), encoding the phosphatase subunit of the trehalose-6-P synthase/phosphatase complex [63] whose deletion was also previously reported to cause 2DG resistance [50]. The identified mutation causes a single substitution (R619C) in the active site of the HAD-like phosphatase domain, which would presumably alter its function, based on structural studies [64].’ Or it could inactivate the enzyme, which causes accumulation of high levels of trehalose-6P, a potent inhibitor of hexokinase.

Line 305 ‘We then studied the phosphorylation of Snf1 and its target Mig1 (Fig 7 A-C). In glucose-grown conditions, the phosphorylation of these proteins was similar in cells expressing WT or mutant alleles of Reg1, in agreement with our hypothesis that Snf1 signaling is not affected (no growth defect, no aberrant DOG2 or SUC2 expression: Fig 4 and 5).’ Does this not indicate that 2DG toxicity is not necessarily related to its glucose repression activity ? Hence, it is important to mention something in the introduction about the presumed connection between the two phenomena.

Line 320 ‘In contrast, the REG1 mutants tested (W165G, A54T, P231S) accumulated 2DG6P to a level that was not significantly different from the WT, suggesting that the lack of 2DG response was not due to an absence of 2DG phosphorylation.’ This is remarkable indeed. There is 2DG phosphorylation but no toxicity. Is it possible that these mutants have higher glucose uptake activity or higher glucokinase activity so that 2DGP is outcompeted by Glu6P ?

Line 325 ‘The phosphorylation of 2DG into 2DG6P is known to be accompanied by a decrease in ATP content, which we observed in reg1∆ cells complemented with WT Reg1 but also with the REG1 mutants tested (Fig 7E).’ Do we have to conclude from this that a drop in ATP or even depletion of ATP does not necessarily compromise growth ?

Line 420 ‘Second, Snf1 is known to inhibit the function of the arrestin-related protein Rod1 in endocytosis [55, 59, 60], and we found that 2DG-induced endocytosis is detrimental to cells in these conditions,’… Endocytosis of what proteins ?

Line 428 ‘First, whereas the reg1∆ strain displays a slow-growth, this was not the case for the missense mutants.’ Slow growth on glucose medium or also on nonfermentable carbon sources ?

Line 450 ‘This also indicates that these Reg1mutants are not resistant because of lack of transport, phosphorylation, or increased detoxification, but rather that 2DG6P is tolerated and does not trigger a response that is detrimental to cell growth in these conditions.’ But there is another important conclusion that the authors overlook: the depletion of ATP caused by the phosphorylation of 2DG and its lack of further metabolism is apparently not a problem for maintenance of cell growth. Or is there no or a reduced drop in ATP in this mutant ? This also raises questions about the intracellular acidification which would normally be caused by ATP depletion because of compromised H+-ATPase activity in the plasma membrane. Would this then not be a problem for cell growth ?

Line 453 ‘This is reminiscent of previous hypotheses that 2DG sensitivity is actually caused by an excessive response’ what type of response is meant ? On Snf1 activity ? Or on something unknown ?

Line 481 ‘except for its ability to confer growth on 2DG.’ What about its ATP level ? When the cells are able to grow one would except at least some level of ATP in the cells.

Line 488 ‘but normal Snf1 activity in glucose-containing medium.’ Please be specific: ‘but normally inhibited Snf1 activity in …’

**Journal Requirements:**

At this stage, the following Authors/Authors require contributions: Alberto Ballin, Veronique Albanese, Samia Miled, Véronique Legros, Guillaume Chevreux, Agathe Verraes, Anne Friedrich, and Sébastien Léon. Please ensure that the full contributions of each author are acknowledged in the "Add/Edit/Remove Authors" section of our submission form.

The list of CRediT author contributions may be found here: https://journals.plos.org/plosgenetics/s/authorship#loc-author-contributions

https://journals.plos.org/plosgenetics/s/submission-guidelines#loc-parts-of-a-submission

5) We notice that your supplementary Figures are included in the manuscript file. Please remove them and upload them with the file type 'Supporting Information'. Please ensure that each Supporting Information file has a legend listed in the manuscript after the references list.

Potential Copyright Issues:

i) Figure 1A. Please confirm whether you drew the images / clip-art within the figure panels by hand. If you did not draw the images, please provide (a) a link to the source of the images or icons and their license / terms of use; or (b) written permission from the copyright holder to publish the images or icons under our CC BY 4.0 license. Alternatively, you may replace the images with open source alternatives. See these open source resources you may use to replace images / clip-art:

7) We note that your Data Availability Statement is currently as follows: "Yes - all data are fully available without restriction. The MS data will be available on PRIDE.". Please confirm at this time whether or not your submission contains all raw data required to replicate the results of your study. Authors must share the “minimal data set” for their submission. PLOS defines the minimal data set to consist of the data required to replicate all study findings reported in the article, as well as related metadata and methods (https://journals.plos.org/plosone/s/data-availability#loc-minimal-data-set-definition).

8) Please amend your detailed Financial Disclosure statement. This is published with the article. It must therefore be completed in full sentences and contain the exact wording you wish to be published.

2) If any authors received a salary from any of your funders, please state which authors and which funders..

9)  Please ensure that the funders and grant numbers match between the Financial Disclosure field and the Funding Information tab in your submission form. Note that the funders must be provided in the same order in both places as well.  

**Have all data underlying the figures and results presented in the manuscript been provided?**

Reviewer #1: Yes

Reviewer #2: **No: ** Is the mass spec data in Fig 9A and 9C available? I may have missed it.

Reviewer #3: Yes

PLOS authors have the option to publish the peer review history of their article (what does this mean? ). If published, this will include your full peer review and any attached files.

**Do you want your identity to be public for this peer review?** For information about this choice, including consent withdrawal, please see our Privacy Policy .

Reviewer #1: No

Reviewer #2: No

Reviewer #3: No

**Figure resubmission:**
---

## [Decision Letter · Decision Letter 1]

29 Sep 2025

Dear Dr. Leon:

We are pleased to inform you that your manuscript entitled "A genetic screen reveals a key role for Reg1 in 2-deoxyglucose sensing and yeast AMPK inhibition" has been editorially accepted for publication in PLOS Genetics. Congratulations!

Yours sincerely,

Anita K. Hopper

Academic Editor

PLOS Genetics

Geraldine Butler

Section Editor

PLOS Genetics

Aimée Dudley

Editor-in-Chief

PLOS Genetics

Anne Goriely

Editor-in-Chief

PLOS Genetics

BlueSky: @plos.bsky.social

Comments from the reviewers (if applicable):

Reviewer's Responses to Questions

**Comments to the Authors:**

Reviewer #1: The authors have addressed my comments and the manuscript has improved.

**Have all data underlying the figures and results presented in the manuscript been provided?**

Reviewer #1: Yes

PLOS authors have the option to publish the peer review history of their article (what does this mean? ). If published, this will include your full peer review and any attached files.

**Do you want your identity to be public for this peer review?** For information about this choice, including consent withdrawal, please see our Privacy Policy .

Reviewer #1: No

**Data Deposition**

http://datadryad.org/submit?journalID=pgenetics&manu=PGENETICS-D-25-00740R1

**Press Queries**

---

## [Editor Report · Acceptance letter]

PGENETICS-D-25-00740R1

A genetic screen reveals a key role for Reg1 in 2-deoxyglucose sensing and yeast AMPK inhibition

Dear Dr Léon,

We are pleased to inform you that your manuscript entitled "A genetic screen reveals a key role for Reg1 in 2-deoxyglucose sensing and yeast AMPK inhibition" has been formally accepted for publication in PLOS Genetics! Your manuscript is now with our production department and you will be notified of the publication date in due course.

With kind regards,

Anita Estes

PLOS Genetics

On behalf of:
